# Dopamine signaling drives skin invasion by human-infective nematodes

Ruhi Patel [1], Gloria Bartolo [1,2], Michelle L. Castelletto[1], Aracely Garcia Romero [1], Astra S. Bryant [3], George W. Agak [4,5] & Elissa A. Hallem [1,5] ✉

Skin-penetrating nematodes are one of the most prevalent causes of disease worldwide. The World Health Organization has targeted these parasites for elimination by 2030, but the lack of preventative measures is a major obstacle to this goal. Infective larvae enter hosts through skin and blocking skin penetration could prevent infection. However, in order to prevent worm ingress via the skin, an understanding of the behavioral and neural mechanisms that drive skin penetration is required. Here, we describe the skin-penetration behavior of the human-infective threadworm *Strongyloides stercoralis*. We show that *S. stercoralis* engages in repeated cycles of pushing, puncturing, and crawling on the skin surface before penetrating. Pharmacological inhibition of dopamine signaling inhibits these behaviors in *S. stercoralis* and the human hookworm *Ancylostoma ceylanicum*, suggesting a critical role for dopamine signaling in driving skin penetration across distantly related nematodes. CRISPR-mediated disruption of dopamine biosynthesis and chemogenetic silencing of dopaminergic neurons also inhibit skin penetration. Finally, inactivation of the TRPN channel TRP-4, which is expressed in the dopaminergic neurons, blocks skin penetration. Our results suggest that drugs targeting TRP-4 and other nematode-specific components of the dopaminergic pathway could be developed into topical prophylactics that block skin penetration, thereby preventing infections.

Skin-penetrating gastrointestinal parasitic nematodes, including the threadworm *Strongyloides stercoralis* and hookworms in the genera *Necator* and *Ancylostoma*, infect over 600 million people worldwide and cause devastating disease and socioeconomic burden[1,2]. Infections by these parasites stunt development in children[3–5], cause chronic disease in both children and adults[1,6,7], and can be fatal for immunocompromised individuals[6,7]. Infections are most prevalent in communities that lack access to sanitation infrastructure and clean drinking water[6,7], which perpetuates a cycle of socioeconomic disparity. Although drug treatments exist, these remedies do not prevent reinfection and may soon be rendered ineffective by the evolution of anthelmintic-resistant nematode populations; indeed, anthelmintic resistance is already a problem among nematodes that parasitize livestock and companion animals[8–14]. Thus, there is an urgent need to expand the existing arsenal of medications to include preventative treatments.

Skin-penetrating nematodes infect hosts by penetrating through host skin[15]. As a crucial step of the infection process, skin penetration is a promising target for intervention – preventative treatments that block skin penetration would stop infections from establishing.

[1]Department of Microbiology, Immunology, and Molecular Genetics, University of California, Los Angeles, Los Angeles, CA, USA. [2]Molecular Biology Interdepartmental PhD Program, University of California, Los Angeles, Los Angeles, CA, USA. [3]Department of Neurobiology and Biophysics, University of Washington, Seattle, WA, USA. [4]Division of Dermatology, Department of Medicine, David Geffen School of Medicine at University of California, Los Angeles, Los Angeles, CA, USA. [5]Molecular Biology Institute, University of California, Los Angeles, Los Angeles, CA, USA. ✉e-mail: ehallem@ucla.edu

Infective larvae are known to penetrate skin head-first[16]; however, beyond this, nothing is known about the behavioral program that is executed during skin invasion. The neural and molecular basis of skin-penetration behavior is also unknown. A mechanistic understanding of this behavior could be harnessed to develop the first prophylactic anthelmintics.

Here, we examine the skin-penetration behavior of the human-infective nematode *S. stercoralis*. We show that infective larvae engage in repeated cycles of pushing on the skin, puncturing the skin, and crawling on the skin before ultimately penetrating the skin. Initial penetration attempts are sometimes aborted; infective larvae then crawl to a new location and re-initiate penetration. Thus, infective larvae actively explore the skin surface before selecting a location to penetrate. Pharmacological inhibition of dopamine signaling impaired skin penetration in *S. stercoralis*, the distantly related human hookworm *Ancylostoma ceylanicum*, and the rat-infective nematode *Strongyloides ratti*, suggesting that dopamine signaling plays a conserved role in driving skin penetration across multiple species of skin-invading nematodes. Skin penetration was also inhibited by CRISPR/Cas9-mediated disruption of the dopamine biosynthesis gene *Sst-cat-2* and chemogenetic silencing of the dopaminergic neurons. Finally, we show that genetic inactivation of the *Sst-trp-4* gene, which encodes a nematode-specific TRPN channel, severely impairs skin penetration. Our results suggest that topical compounds that block *Sst*-TRP-4 or another nematode-specific component of the dopaminergic pathway could function as the first topical repellents for skin-penetrating nematodes.

## Results

### *S. stercoralis* infective larvae engage in repeated behavioral cycles on skin

*S. stercoralis* penetrates human skin as developmentally arrested, infective third-stage larvae (iL3s)[17]. After host invasion, development resumes and the nematodes follow a complicated life cycle that includes both intra-host and extra-host life stages[17,18] (Supplementary Fig. S1). To examine the behavior of *S. stercoralis* iL3s on skin, we developed an ex vivo tracking assay that enabled us to observe and quantify skin penetration in real time (Fig. 1A). Briefly, we excised the skin from euthanized rats, sectioned the skin, removed the fur, and then suspended skin pieces over saline using plastic inserts. We then placed individual, fluorescent *S. stercoralis* iL3s on the surface of the skin and used a fluorescence dissection microscope and attached camera to acquire time-lapse images of worm behavior. The *S. stercoralis* iL3s were fluorescent across the entire body either because of stable expression of an *Sst-act-2p::strmScarlet-I* reporter cassette[19] or because they were labeled with DiI (1,1′-dioctadecyl-3,3,3′,3′-tetramethylindocarbocyanine perchlorate), which stains the nematode cuticle[20]. This allowed visualization of the translucent worms on the skin surface. We note that iL3s stained with DiI moved at a slightly reduced speed relative to unstained iL3s but otherwise exhibited similar behaviors on skin (Supplementary Fig. S2A-B). In addition, skin explants used for ex vivo assays were frozen and then thawed shortly before the experiment; we did not observe significant differences between the proportion of iL3s that penetrated fresh vs. frozen skin (Supplementary Fig. S2C).

When *S. stercoralis* iL3s were placed on rat skin, they typically crawled a short distance (Fig. 1B, C, Supplementary Movies S1, S2) and then halted forward locomotion and pushed down perpendicularly on the skin surface with their heads (Fig. 1B, C, Supplementary Movies S1, S2). These pushes, where the head of the iL3 indented but did not pierce the skin (Fig. 1B, C), appeared to be a means of "sampling" the skin surface to identify soft spots or openings such as hair follicles that could be exploited for invasion. During a push, the entire body of the worm was always detectable on the skin surface (Fig. 1C, Supplementary Movies S1, S2). After pushing, iL3s either

crawled to a distinct spot or initiated a skin-penetration attempt by puncturing the skin with their heads (Fig. 1B, D, Supplementary Movies S1, S2). A puncture event was scored if either of the following two conditions were met: (1) the nose of the worm was detected inside the skin, as indicated by a decrease in fluorescence and blurring of the nose, or (2) the nose had disappeared from the skin surface while the worm moved into the skin (Fig. 1D, E, Supplementary Movies S1, S2). Moreover, a worm that had punctured the skin appeared anchored to the skin surface by its head while the rest of its body moved freely. Once a penetration attempt was initiated by a puncture, iL3s either continually burrowed into the skin until penetration was completed (i.e., the full body of the iL3 was inside the skin) (Fig. 1B, D), or they retreated to the skin surface and aborted the penetration attempt (Fig. 1B, E). Worms aborted penetration attempts either by reversing out of the skin or by executing a turn within the skin and crawling out in the forward direction. Aborted penetration attempts were rare, as only 1 out of the 17 iL3s observed in our assay showed aborted penetration attempts; that iL3 aborted 2 out of 3 attempts (Fig. 2A, bottom track on rat skin). Such aborted penetration attempts may occur when iL3s encounter a structure within the skin that provides mechanical resistance to entry, such as the basement membrane, sebaceous glands, or fibroblasts[21]. The iL3 that aborted penetration attempts subsequently re-initiated and completed penetration at a distinct site (Fig. 2A). Thus, skin penetration involves repeated sampling of the skin surface through cycles of pushing, puncturing, and crawling until the iL3 ultimately completes penetration. The finding that iL3s actively explore the skin surface to locate favorable entry points, rather than diving into the skin immediately upon contact, suggests that topical compounds that interfere with these behaviors could be developed into novel anti-nematode prophylactics.

### Skin-penetration behavior increases on host skin

We next asked whether *S. stercoralis* skin-penetration behavior is enhanced on host vs. non-host skin. To answer this question, we examined the behavior of *S. stercoralis* iL3s on human skin; skin samples were obtained either from the forearm of cadavers or from the abdomen or breast of surgical patients following plastic surgery (Supplementary Fig. S3). We found that *S. stercoralis* iL3s pushed down on human skin more frequently than rat skin (Fig. 2A, B). *S. stercoralis* iL3s also pushed down more rapidly after placement on human skin than rat skin (Fig. 2C). Together, these observations suggest that *S. stercoralis* iL3s exhibit more skin-penetration behavior on the skin of a definitive host, humans, as compared with non-host rat skin. Nevertheless, *S. stercoralis* iL3s were equally able to penetrate both skin types (Fig. 2D), indicating that factors besides the frequency of pushes and punctures, such as skin toughness, likely influence success at skin penetration.

We next examined whether the rat-infective nematode *S. ratti* executed similar behaviors to *S. stercoralis* during skin penetration, and whether these behaviors were also enhanced on host skin. We found that like *S. stercoralis* iL3s, *S. ratti* iL3s repeatedly pushed and punctured rat skin until penetration was complete (Supplementary Movie S3, Fig. 2E). Overall, *S. ratti* iL3s pushed and punctured host rat skin more frequently than non-host human skin (Fig. 2E, F). However, *S. ratti* iL3s executed the first push at roughly the same time on both rat and human skin (Fig. 2E, G). Thus, while some aspects of *S. ratti* skin-penetration behavior were enhanced on host skin, other aspects were similar on both skin types. Ultimately, 100% of *S. ratti* iL3s penetrated host rat skin, whereas only ~40% penetrated non-host human skin. This difference in penetration ability might be partially attributable to the fact that rat skin is much softer than human skin[22]. Based on our findings, we suggest that skin-penetration behavior and ability are likely an interplay between worm-specific factors (*e.g.*, the ability of the worm to detect skin-

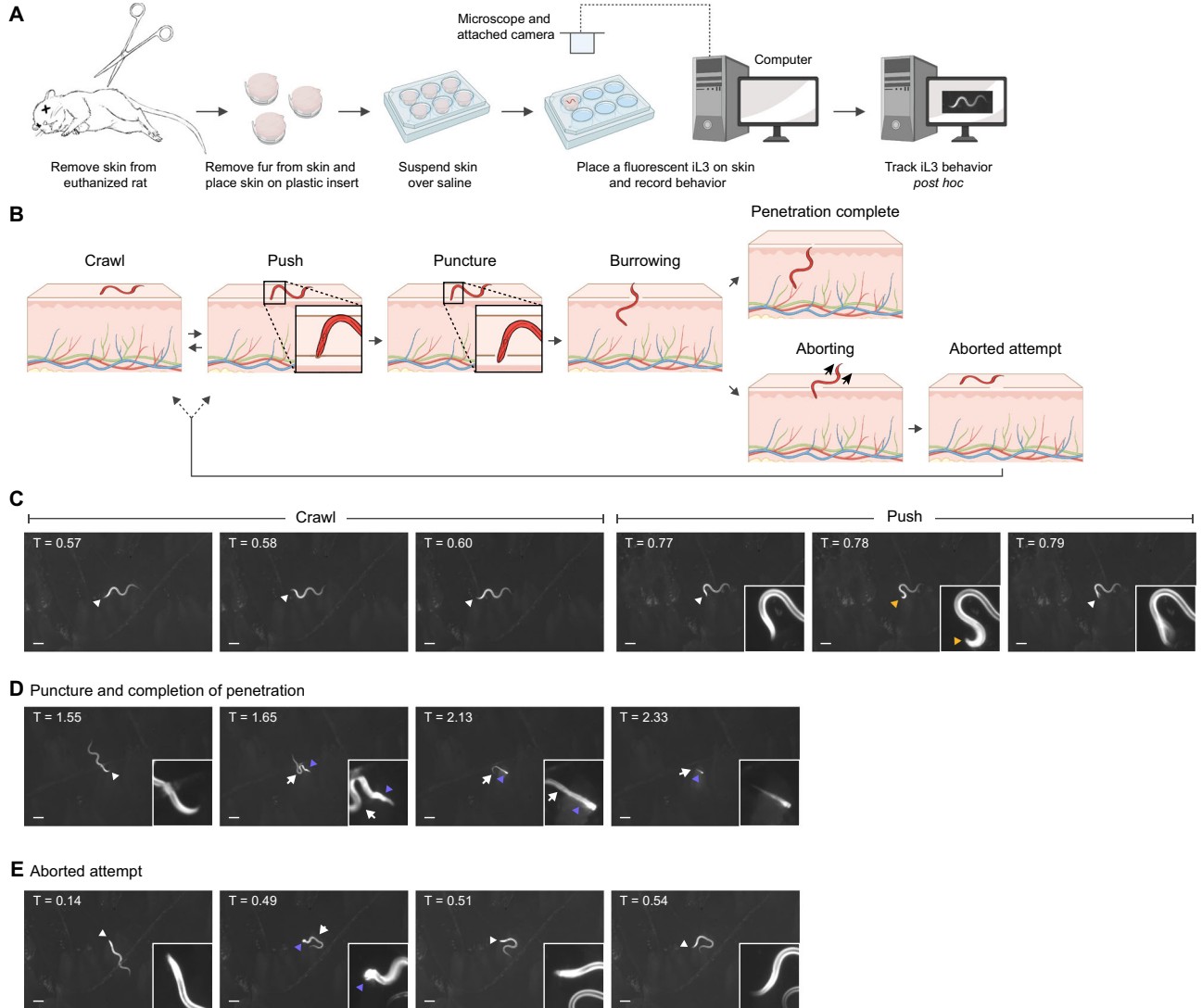

**Fig. 1 | *S. stercoralis* iL3s engage in repeated behavioral motifs on skin. A** An ex vivo assay for quantifying behavior on skin. Rat skin is excised, epilated, sectioned, and suspended over BU saline[79]. Next, fluorescent iL3s are placed on the skin surface and time-lapse images are acquired for up to 5 min. **B** Schematic depicts the behaviors executed by iL3s on skin. Pushes are characterized by pauses in locomotion, coupled with the worm moving its head against the skin without piercing it (inset). Punctures occur when the head breaches the skin surface (inset). Thereafter, the worm either burrows into the skin until penetration is complete or aborts the penetration attempt and returns to the surface. Created in BioRender. Mushtaqh Ali, R. (2025) https://BioRender.com/79cx30x. **C–E** Time-lapse images of an *S. stercoralis* iL3 on rat skin. See also Supplementary Movie S1. **C** The iL3 crawled on the skin surface (left three panels) and pushed against the skin (right three panels). The head is blurred in the rightmost panel because the worm was actively pushing against the skin. **D** The same iL3 puncturing and penetrating skin. The first

panel shows the iL3 outside the skin; the second panel shows that the iL3 has punctured the skin; the third panel shows the iL3 burrowing into the skin; and the fourth panel shows that the iL3 has almost completed skin penetration, with only the tip of the tail outside. **E** The same iL3 aborting an earlier penetration attempt. The first panel shows the iL3 prior to puncturing the skin and the second panel shows that the iL3 has punctured the skin, with its head no longer detectable. The third and fourth panels show that the iL3 has aborted the penetration attempt and returned to the surface. In panels (**C–E**), white arrowheads indicate the head of the worm, yellow arrowheads indicate pushing, purple arrowheads indicate the skin entry point, and white arrows indicate the part of the worm outside the skin. Timestamps (min) for each panel are relative to the time of placement on skin. Scale bar = 100 μm. Insets are magnified 4× and the contrast is increased. **A** and **B** were generated in BioRender.

specific sensory cues) and skin-specific factors (*e.g.*, mechanical properties of the skin such as stiffness and microstructure).

### Pharmacological inhibition of dopamine receptors inhibits skin penetration in *Strongyloides* species and human hookworms

We next investigated the neural basis of skin-penetration behavior. Our behavioral analysis suggested that iL3s survey the texture of the skin surface in search of favorable entry points. In the free-living nematode *Caenorhabditis elegans*, the dopaminergic neurons mediate detection of textured surfaces[23–26]. We therefore hypothesized that dopamine signaling might regulate the detection of skin surface

texture and drive the ensuing behavioral response in *S. stercoralis*. To test this, we treated *S. stercoralis* iL3s with haloperidol, which interferes with the activity of dopamine receptors[27,28], and performed single-worm skin-penetration assays on rat skin explants. We found that haloperidol-treated iL3s crawled on the skin in circuitous paths, rarely pushed or punctured the skin, and often failed to penetrate (Fig. 3A–D). These behavioral phenotypes were rescued by the addition of exogenous dopamine (Fig. 3A–D), suggesting that haloperidol acts on the dopaminergic pathway to modulate skin penetration. Similar results were observed upon treatment of *S. ratti* iL3s with haloperidol and dopamine (Supplementary Figs. S4, S5).

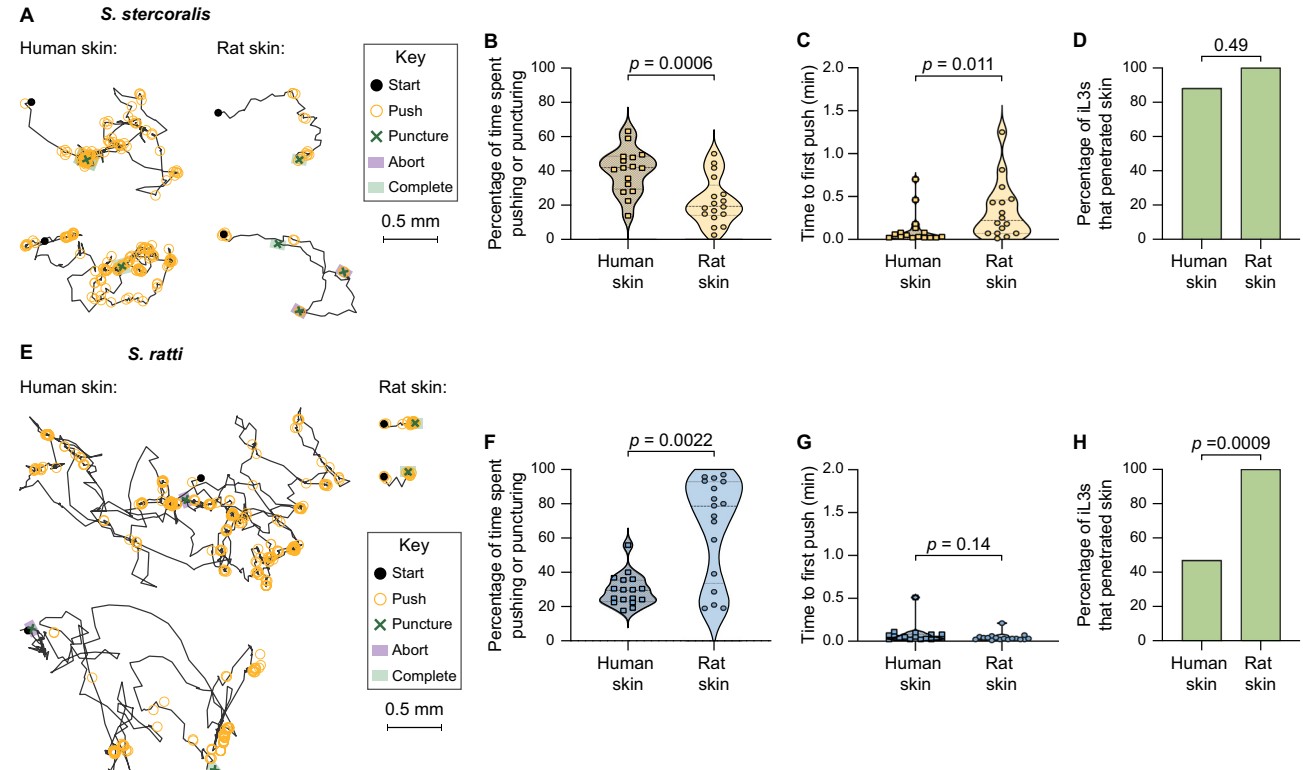

**Fig. 2 | Skin-penetration behavior is conserved across species but occurs more frequently on host skin. A** *S. stercoralis* iL3s show more skin-penetration behavior on human than rat skin. Two representative iL3s on each skin type are shown; key shows behavioral motifs that were tracked. **B** Violin plot shows the percentage of time *S. stercoralis* iL3s pushed and punctured the surface of human or rat skin. *n* = 16 and 17 iL3s on human and rat skin, respectively. **C** Violin plot shows the time taken by *S. stercoralis* iL3s to push for the first time since placement on either human or rat skin. *n* = 16 iL3s per skin type. **D** Bar graph shows the percentage of *S. stercoralis* iL3s that penetrated human or rat skin. *n* = 17 iL3s per skin type. **E** *S. ratti* iL3s push down on rat skin more than human skin. Two representative iL3s on each skin type are shown. Key shows behavioral motifs. **F** Violin plot shows the percentage of time *S. ratti* iL3s spent pushing or puncturing the surface of human or rat

skin. *n* = 17 iL3s per skin type. **G** Violin plot shows the time taken by *S. ratti* iL3s to push down for the first time since placement on either human or rat skin. *n* = 17 iL3s per skin type. **H** Bar graph shows the percentage of *S. ratti* iL3s that penetrated human or rat skin. *n* = 17 iL3s per skin type. For (**B**, **C**, **F**, **G**), dots, dashed lines, and dotted lines indicate individual worms, medians, and interquartile ranges, respectively. Data in (**B**–**D**) were obtained from 4 independent replicates, and data in (**F**–**H**) from 3 independent replicates. Skin from three human donors was tested. A two-tailed unpaired *t*-test was used in (**B**), two-tailed Mann–Whitney tests were used in (**C**, **F**, **G**), and two-tailed Fisher's exact tests were used in (**D**, **H**). There are no error bars in (**D**, **H**) because graphs show percentages of the total iL3s tested. Transgenic EAH435 and EAH414 strains were used in these experiments. Source data are provided in the Source Data file.

We next tested whether haloperidol treatment could prevent iL3s from penetrating the skin of a live host, by treating *S. ratti* iL3s with haloperidol and performing skin-penetration assays on live rats. Briefly, we anesthetized live rats, removed the fur from discrete skin sections, placed haloperidol-treated or control iL3s on these fur-free sections, and observed skin-penetration behavior for 5 min thereafter or until the iL3 either penetrated the skin or stopped moving on the skin surface. We found that haloperidol treatment dramatically reduced skin penetration on live rats: only ~10–20% of the haloperidol-treated iL3s penetrated live rat skin, compared to ~80% of the control iL3s (Fig. 3E). Most haloperidol-treated iL3s also failed to puncture the skin (Fig. 3F), indicating that treatment of iL3s with haloperidol blocks the initiation of skin penetration. These results suggest a critical role for dopamine signaling in regulating skin penetration in vivo.

To test whether dopamine signaling also regulates skin penetration in other human-infective nematodes, we repeated these experiments with the distantly related human-parasitic hookworm *Ancylostoma ceylanicum*. We found that treatment of *A. ceylanicum* iL3s with haloperidol reduced pushes and punctures and inhibited penetration of rat skin (Fig. 4A–C). Haloperidol-treated *A. ceylanicum* iL3s that failed to penetrate the skin also often failed to puncture the skin (Fig. 4D), showing that blocking dopamine signaling reduces skin-penetration behavior. Dopamine rescued these behavioral defects (Fig. 4A–D). Together, our results demonstrate that dopamine

signaling plays a conserved role in driving skin penetration across distantly related nematode species and highlight the potential for drugs that target the dopaminergic pathway to serve as broad-spectrum anti-nematode prophylactics.

## Disrupting dopamine biosynthesis blocks skin penetration

To directly test the role of dopamine signaling during skin penetration, we disrupted dopamine biosynthesis using CRISPR/Cas9-mediated targeted mutagenesis. We first mined the *S. stercoralis* genome and identified a putative ortholog of the *C. elegans* gene *Cel-cat-2*, which encodes a tyrosine hydroxylase that mediates dopamine biosynthesis[29,30] (Fig. 5A). In parallel, we also identified a putative ortholog of the gene encoding the *C. elegans* dopamine transporter, *Cel-dat-1*[31] (Supplementary Fig. S6A). The amino acid sequences of *Sst*-CAT-2 and *Cel*-CAT−2 were 44% identical overall and 56% identical in the predicted catalytic domain[32] (Supplementary Fig. S6B). Similarly, *Sst*-DAT-1 and *Cel*-DAT-1 were 59.5% identical overall and 69.4% identical in the predicted monoamine transporter domain (Supplementary Fig. S6C); the monoamine transporter domains were predicted using the conserved domain database (CDD)[32]. Transcriptional reporters for *Sst-cat-2* and *Sst-dat-1* were co-expressed in several cells that occupy the same position in iL3s as the *C. elegans* dopaminergic neurons[33,34] (Fig. 5B, Supplementary Fig. S7, Supplementary Movie S4). Specifically, the transcriptional reporters were expressed in a set of cells that send

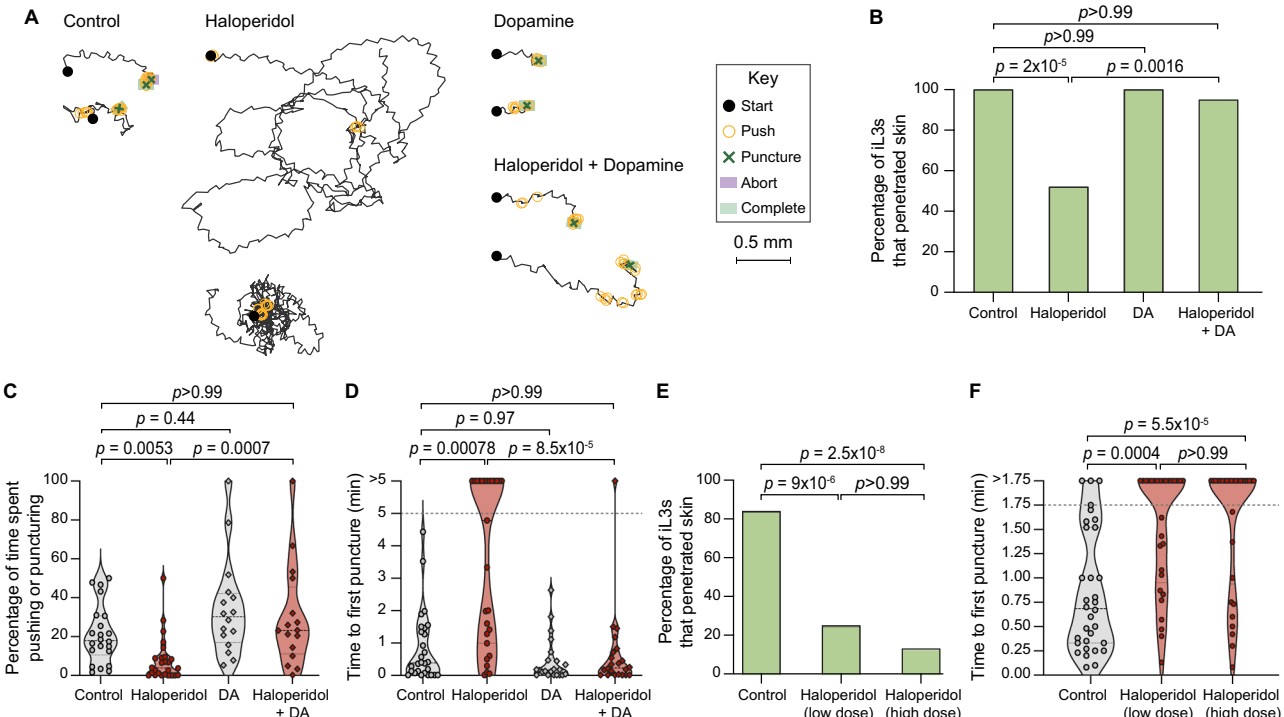

**Fig. 3 | Pharmacological inhibition of dopamine signaling blocks skin penetration in *Strongyloides*. A** Haloperidol inhibits penetration and dopamine rescues this phenotype. Tracks show representative *S. stercoralis* iL3s from each treatment group. Key shows behavioral motifs. **B** Graph shows the percentage of *S. stercoralis* iL3s treated with vehicle ("control"), 1.5 mM haloperidol, 10 mM dopamine (DA), or 1.5 mM haloperidol +10 mM DA that completed penetration. *n* = 28 control, 27 haloperidol-treated, 22 DA-treated and 22 haloperidol+DA-treated iL3s. **C** Violin plot depicts the percentage of time *S. stercoralis* iL3s pushed and punctured skin. *n* = 22 control, 26 haloperidol-treated, 16 DA-treated, and 16 haloperidol +DA-treated iL3s. **D** Violin plot depicts the time taken by *S. stercoralis* iL3s to first puncture skin. Dotted line at y = 5 indicates the assay end time; dots above this line indicate animals that failed to puncture. *n* = 28 control, 27 haloperidol-treated, 22 DA-treated, and 22 haloperidol+DA-treated iL3s. **E** Graph shows the percentage of

*S. ratti* iL3s treated with vehicle ("control"), low-dose haloperidol (160 µM), or high-dose haloperidol (1 mM) that penetrated live rat skin. *n* = 32 iL3s per condition. **F** Violin plot depicts the time taken by *S. ratti* iL3s to first puncture live rat skin. Worms that did not puncture the skin by 1.75 min were set to ">1.75". *n* = 32 control, 29 160-µM-treated, and 30 1-mM-treated iL3s. For violin plots, dots, dashed lines, and dotted lines depict individual worms, medians, and interquartile ranges, respectively. Data in (**B**–**D**) and (**E**, **F**) are from 4 independent replicates. Data in (**E**, **F**) are from 7 male and 2 female rats. Two-tailed Fisher's exact tests with Bonferroni's correction were used in (**B**, **E**), and Kruskal–Wallis tests with Dunn's post-test were used in (**C**, **D**, **F**). There are no error bars in (**B**, **E**) because graphs show percentages of the total iL3s tested. DiI-stained iL3s and rat skin were used in these experiments. Source data are provided in the Source Data file.

processes to the tip of the nose, which are likely the *Sst*-CEP neurons (Fig. 5B, Supplementary Fig. S7); cells immediately posterior to the candidate *Sst*-CEP neurons that are likely the *Sst*-ADE neurons (Fig. 5B, Supplementary Fig. S7); and cells along the body of the iL3 that send processes along the ventral side of the worm and are likely the *Sst*-PDE neurons (Fig. 5B, Supplementary Fig. S7)[33,34]. The conserved positions of the dopaminergic neurons in *S. stercoralis* and *C. elegans* is consistent with the general conservation of sensory neuroanatomy and function between the two species[35–41]. Together, the phylogenetic analysis and spatial expression profile of *Sst-cat-2* suggest that it is indeed the *S. stercoralis* tyrosine hydroxylase gene. Both *Sst-cat-2* and *Sst-dat-1* are significantly upregulated in iL3s relative to other life stages (Fig. 5C, D)[42,43], consistent with a critical role for dopamine signaling in mediating iL3-specific behaviors.

We next disrupted the *Sst-cat-2* gene using CRISPR and then generated a stable mutant line by propagation of homozygous mutants in Mongolian gerbils, the laboratory host for *S. stercoralis*, as previously described[41] (Fig. 5E, Supplementary Figs. S8, S9). We found that inactivation of *Sst-cat-2* nearly eliminated skin-penetration behavior (Fig. 6A-B, Supplementary Fig. S10A, B). Instead, *Sst-cat-2*[−/−] iL3s crawled in long, circuitous paths on the skin surface and rarely engaged in pushes (Fig. 6A, C; Supplementary Fig. S10A, C). When they did push on the skin, the pushing bouts were ~2-fold shorter in *Sst-cat-2*[−/−] iL3s relative to wild-type iL3s (Fig. 6D, Supplementary Fig. S10D). Thus, inactivation of *Sst-cat-2* reduces the propensity of iL3s to halt

locomotion and engage in focused pushing bouts. Nearly half of the *Sst-cat-2*[−/−] iL3s failed to puncture rat skin (Fig. 6E), and two-thirds failed to puncture human skin (Supplementary Fig. S10E); the less severe inhibition of punctures on non-host rat skin might be because rat skin is softer[22] and therefore likely easier to puncture than human skin. Interestingly, *Sst-cat-2*[−/−] iL3s also frequently reversed after a push or puncture, whereas wild-type iL3s did not (Fig. 6F, Supplementary Fig. S10F); this likely explains why the majority of the mutants that did puncture the skin subsequently backed out of the skin and aborted the penetration attempt (Fig. 6G). Together, our results indicate that dopamine signaling controls multiple facets of skin-penetration behavior: pushes, punctures, and the ability to successfully complete penetration following a puncture.

## Dopaminergic neurons drive skin penetration

To further examine the role of dopamine signaling in mediating skin penetration, we chemogenetically silenced the dopaminergic neurons using the histamine-gated chloride channel HisCl1[44] (Fig. 7A, Supplementary Fig. S11A). We found that like *Sst-cat-2*[−/−] iL3s, iL3s with silenced dopaminergic neurons rarely engaged in pushes and punctures and instead actively crawled across the skin surface (Fig. 7B–D). More than half of the iL3s failed to penetrate the skin (Fig.7C) and this was often because of a failure to puncture the skin (Fig. 7E). These results indicate that the dopaminergic neurons of *S. stercoralis* drive skin-penetration behavior. Histamine treatment of

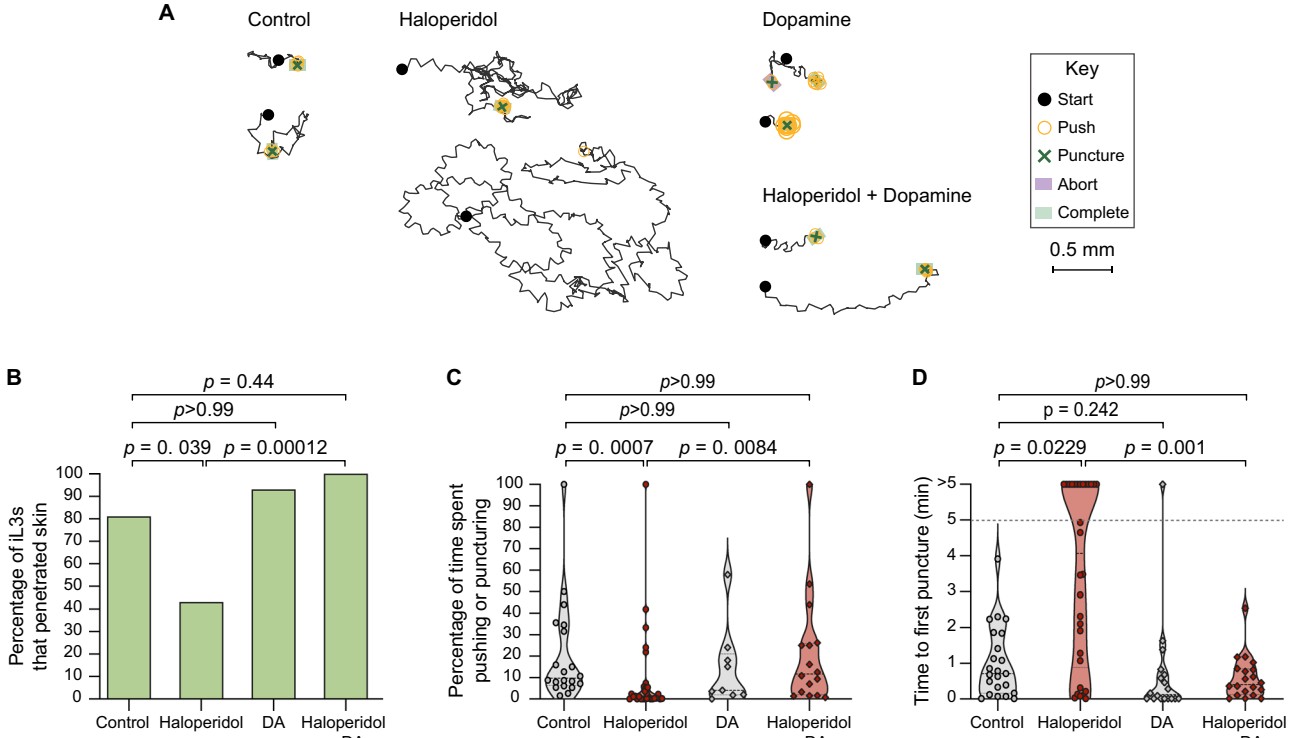

**Fig. 4 | Pharmacological inhibition of dopamine signaling blocks skin penetration in the human-parasitic hookworm *A. ceylanicum*. A** Haloperidol inhibits skin penetration, and dopamine rescues this phenotype. Tracks show two representative worms from each treatment group; for the haloperidol-treated group, one representative worm that completed penetration and one randomly selected worm that did not complete penetration are shown. Key shows behavioral motifs. **B** Bar graph shows the percentage of *A. ceylanicum* iL3s that were treated with either the vehicle ("control"), 160 μM haloperidol, 10 mM DA, or 160 μM haloperidol +10 mM DA that completed penetration. *n* = 21 control, 31 haloperidol-treated, 14 DA-treated, and 18 haloperidol+DA-treated iL3s. There are no error bars because the graph shows the percentage of iL3s in each treatment group that completed penetration out of the total number tested. **C** Violin plot depicts the percentage of

time that *A. ceylanicum* iL3s spent engaging in pushes or punctures. *n* = 20 control, 29 haloperidol-treated, 9 DA-treated, and 16 haloperidol+DA-treated iL3s. **D** Violin plot depicts the time taken by *A. ceylanicum* iL3s to first puncture skin. The dotted line at y = 5 indicates the time at which the assay ended; the dots above this line indicate worms that failed to puncture. *n* = 23 control, 30 haloperidol-treated, 18 DA-treated, and 20 haloperidol+DA-treated iL3s. For (**C**, **D**), dots depict individual worms, dashed lines indicate medians, and dotted lines indicate interquartile ranges. Data in (**B**–**D**) were obtained from 4 independent replicates. A two-tailed Fisher's exact test with Bonferroni correction for multiple comparisons was used in (**B**), and Kruskal–Wallis tests with Dunn's post-test were used in (**C**, **D**). DiI-stained iL3s and rat skin were used in these experiments. Source data are provided in the Source Data file.

wild-type iL3s that did not express the *HisCl1* transgene did not significantly alter skin-penetration behavior, indicating that the behavioral phenotypes described above are specific to silencing of the dopaminergic neurons (Supplementary Fig. S11B–D). We note that HisCl1-mediated silencing may not result in complete loss of dopaminergic neuron activity, which is likely why the phenotype of the iL3s with silenced dopaminergic neurons was slightly less severe than that of the *Sst-cat-2*[−/−] iL3s.

### *Sst*-TRP-4 represents a possible target for topical prophylactic intervention

We next asked whether there are nematode-specific components of the dopaminergic pathway that could be targeted for nematode control without interfering with host dopamine signaling. We hypothesized that the transient receptor potential channel TRP-4 might be one such target. In *C. elegans*, *Cel*-TRP-4 is expressed in the dopaminergic neurons and couples mechanosensation of textured surfaces (*e.g.*, bacterial lawns) with behavioral responses[24,25,45]. We therefore asked whether TRP-4 is conserved in *S. stercoralis* and if so, whether it might have been co-opted in the *S. stercoralis* dopaminergic neurons to drive skin-penetration behavior. We identified a putative ortholog of *Cel*-TRP-4 in the *S. stercoralis* genome (Fig. 8A); the amino acid sequences of *Sst*-TRP-4 and *Cel*-TRP-4 were 57.1% identical (Supplementary Fig. S12). An *Sst-trp-4* transcriptional reporter was co-expressed with *Sst-dat-1* in the putative *Sst*-CEP and *Sst*-ADE neurons (Fig. 8B), and

expression of the *Sst-trp-4* gene was significantly upregulated in the iL3 stage relative to other life stages (Fig. 8C)[42,43]. Taken together, these findings indicate that *Sst-trp-4* is expressed in the *S. stercoralis* dopaminergic neurons and might play an important role in iL3-specific behaviors.

To test whether *Sst-trp-4* is necessary for skin penetration, we generated a stable *Sst-trp-4*[−/−] knockout line[41], as described above (Fig. 8D, Supplementary Figs. S8, S13). We then performed skin-penetration assays with *Sst-trp-4*[−/−] and wild-type iL3s on both rat and human skin. Similar to inactivation of *Sst-cat-2*, inactivation of *Sst-trp-4* drastically reduced skin penetration (Fig. 9A, B, Supplementary Fig. S14A, B). *Sst-trp-4*[−/−] iL3s pushed on the skin less frequently than control iL3s (Fig. 9A, C; Supplementary Fig. S14A, C), and the pushing bouts that did occur were shorter and more often followed by a reversal (Fig. 9D, F; Supplementary Fig. S14D). Moreover, over 60% of the mutants on rat skin and 50% of the mutants on human skin failed to puncture the skin at all (Fig. 9E, Supplementary Fig. S14E), and the penetration attempts that were initiated were often aborted (Fig. 9G, Supplementary Fig. S14F). These results show that *Sst*-TRP-4 plays a key role in driving skin penetration in *S. stercoralis*. Notably, *Sst*-TRP-4 is not conserved to humans[46] (Supplementary Fig. S15) but is conserved to the human-infective hookworm *A. ceylanicum* (Supplementary Fig. S16). Thus, topical compounds containing drugs that target *Sst*-TRP-4 have the potential to prevent infections by multiple species of skin-penetrating nematodes while causing little to no side effects in humans.

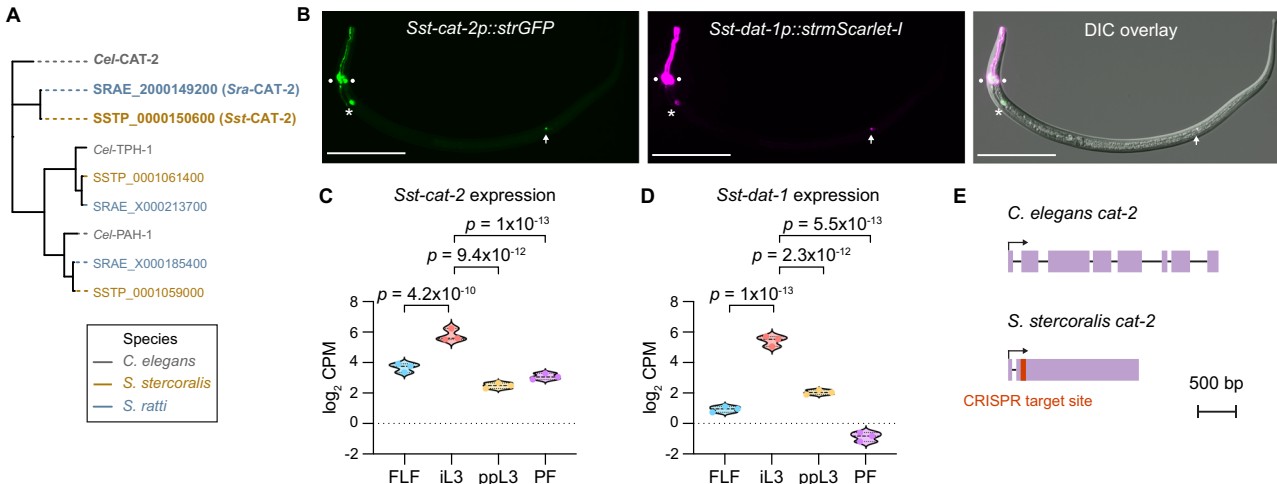

**Fig. 5 | Identification of *S. stercoralis cat-2*. A** Phylogenetic analysis shows the closest homologs of *C. elegans* CAT-2 (gray) in *S. stercoralis* (brown) and *S. ratti* (blue). **B** Co-expression of *Sst-cat-2* and *Sst-dat-1* in the putative dopaminergic neurons of *S. stercoralis*. Montage shows expression of an *Sst-cat-2* transcriptional reporter (green), expression of an *Sst-dat-1* transcriptional reporter (magenta), and co-expression of the two reporters and the relative positions of these neurons along the body of the iL3 in the differential interference contrast (DIC) overlay. The *Sst-cat-2* transcriptional reporter comprises a 1639 bp region upstream of the *Sst-cat-2* start codon fused with a gene encoding GFP that is codon-optimized for expression in *Strongyloides* species (*strGFP*). Similarly, the *Sst-dat-1* transcriptional reporter comprises a 2579 bp region upstream of the *Sst-dat-1* start codon fused with *Strongyloides*-codon-optimized *mScarlet-I* (*strmScarlet-I*). The circles, asterisk, and arrow label the putative *Sst*-CEP, *Sst*-ADE, and *Sst*-PDE neurons, respectively.

Dorsal is up; head is left. Scale bar = 100 μm. **C, D** Violin plots show expression levels of *Sst-cat-2* and *Sst-dat-1*, in log₂ counts per million (CPM), at the indicated life stages based on published RNA-seq datasets[42,43,68]. Statistical tests for differential gene expression analysis were performed using the limma R package as previously described[42], and *p*-values were adjusted for multiple comparisons using the Benjamini-Hochberg method[42]. FLF free-living female, iL3 infective third-stage larva, ppL3 post-parasitic third-stage larva, PF parasitic female. Each dot indicates an independent replicate. **E** The *cat-2* genes of *C. elegans* and *S. stercoralis*. Schematics show the gene models of *Cel-cat-2* (isoform a) and *Sst-cat−2*. Exons and introns are depicted as lavender boxes and black lines, respectively. Transcriptional start sites are indicated by arrows. *Sst-cat-2* has a single CRISPR/Cas9 target site in the second exon (red). Drawings are to scale and scale bar = 500 bp. Source data are provided in the Source Data file.

## Discussion

Here, we present an in-depth analysis of skin penetration, a critical but previously unstudied behavior that enables skin-penetrating nematodes to invade mammalian hosts. We show that iL3s execute a complex set of behavioral motifs that allows them to probe the skin surface for favorable points of entry and culminates in the penetration of skin tissue (Fig. 1). Favorable entry points could include skin surface contours that enable iL3s to gain traction for burrowing into the skin, gaps between individual epidermal keratinocytes that provide reduced resistance to burrowing, hair follicles, and open wounds. The finding that infective larvae actively explore the skin surface to locate favorable entry points, rather than entering the skin immediately upon contact, identifies a window of opportunity for preventative interventions and suggests that topical compounds that interfere with skin-penetration behaviors could be developed into the first anti-nematode prophylactic treatments.

We also uncover neural and molecular mechanisms that underlie skin penetration. By treating worms with haloperidol, a drug that interferes with the activity of the dopamine receptors, we show that dopamine signaling is necessary for skin penetration by *S. stercoralis*, *S. ratti*, and *A. ceylanicum* (Figs. 3 and 4 and Supplementary Fig. S4). *S. stercoralis* and *A. ceylanicum* occupy distinct phylogenetic clades, and they are thought to have evolved independently from free-living ancestors[47]. Thus, our findings imply that the role of dopamine signaling in driving skin penetration evolved independently in the two species of parasitic nematodes. Going further, we show that the dopamine biosynthesis enzyme *Sst*-CAT-2, the dopaminergic neurons, and the putative mechanoreceptor *Sst*-TRP-4 mediate skin-penetration behavior in *S. stercoralis* (Figs. 5–9; Supplementary Figs. S10, S14). *Sst*-TRP-4 is conserved to *A. ceylanicum* but not to humans (Supplementary Figs. S15, 16), suggesting that preventative interventions that target this protein may be both broadly effective against diverse species of skin-penetrating nematodes and safe for administration to humans.

However, we note that hookworms are also capable of oral infection[48,49], and thus drugs that prevent skin penetration may be more effective in preventing infections with *Strongyloides* species than hookworms.

Skin penetration is a parasite-specific behavior. Sensory neuroanatomy is largely conserved across nematode species[35,37], raising the question of how parasite-specific behaviors have evolved in parasitic nematodes despite their having essentially the same sensory neuroanatomy as free-living nematodes. In *C. elegans*, the dopaminergic neurons are necessary for the worms to slow down upon entry into a patch of food, and treatment of *C. elegans* with exogenous dopamine halts locomotion[23,25,50]. In skin-penetrating nematodes, the dopaminergic neurons have been co-opted to drive skin penetration. Our data suggest that dopamine signaling causes the worm to stop crawling and instead push its head against the skin. Indeed, we show that the average duration of pushing bouts in the *Sst-cat-2*⁻/⁻ mutants is shorter than wild type (Fig. 6D, Supplementary Fig. S10D), indicating that *Sst-cat-2*⁻/⁻ iL3s have a reduced propensity to halt locomotion and push down on skin. Thus, while dopamine signaling appears to cause pauses in locomotion in both free-living and skin-penetrating nematodes, in skin-penetrating nematodes the dopaminergic neurons are coupled to an additional downstream motor program that stimulates pushing against the skin and ultimately skin penetration.

The exact role of dopamine signaling and the dopaminergic neurons in skin penetration is not yet fully characterized. Based on our results, we propose a model whereby the dopaminergic neurons sense topographical features of the skin surface, via the mechanotransduction channel *Sst*-TRP-4, and then release dopamine. Dopamine binds to downstream dopamine receptors, causing the worm to slow forward crawling and instead push against the skin; pushing behavior likely enables worms to identify entry points into the skin. Consistent with the model that dopamine and *Sst*-TRP-4 are necessary for pushes, we show that *Sst-cat-2*⁻/⁻ and *Sst-trp-4*⁻/⁻ iL3s push and puncture skin for

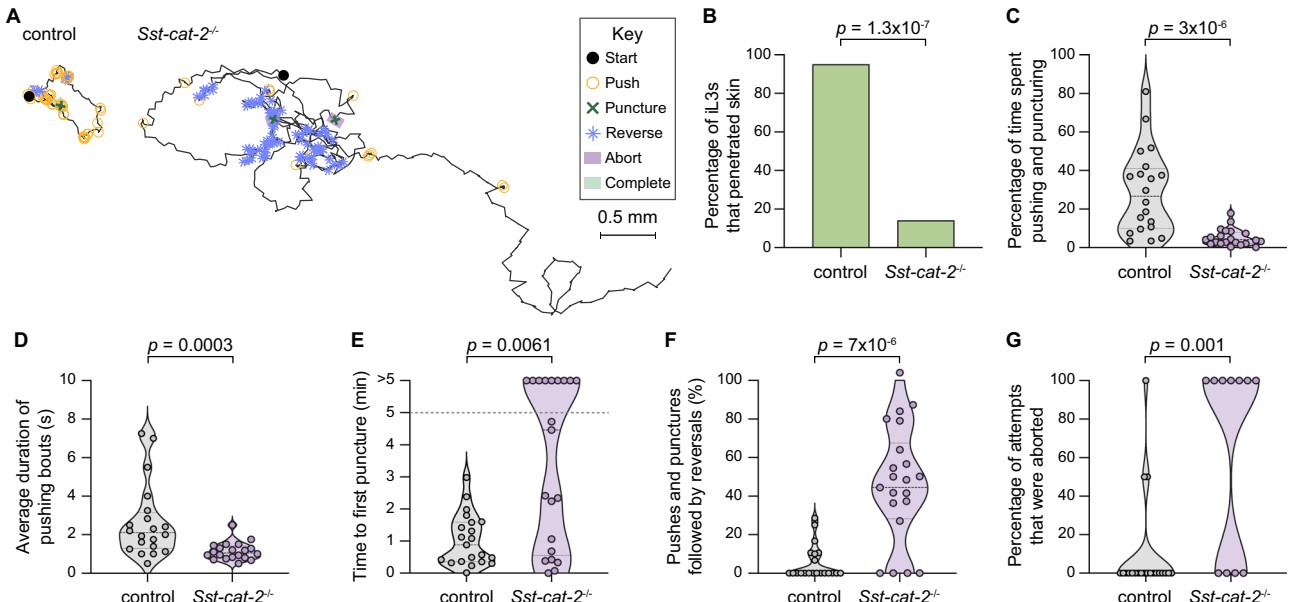

**Fig. 6 | Dopamine is required for skin penetration. A** Inactivation of *Sst-cat-2* drastically alters skin-penetration behavior on rat skin. Tracks show the skin-penetration behaviors of a representative control iL3 that punctured and penetrated the skin and a representative *Sst-cat-2⁻/⁻* iL3 that punctured but did not complete penetration. Key shows behavioral motifs. **B** Bar graphs show the percentage of control and *Sst-cat-2⁻/⁻* iL3s that completed skin penetration. *n* = 21 iL3s per genotype. There are no error bars in the bar graph because the graph shows the percentage of iL3s of each genotype that completed penetration out of the total number tested. **C** Violin plot depicts the percentage of time on skin that control and *Sst-cat-2⁻/⁻* iL3s spent engaging in pushes or punctures. *n* = 20 control and 21 *Sst-cat-2⁻/⁻* iL3s. **D** Violin plot shows the average duration of pushing bouts for control vs. *Sst-cat-2⁻/⁻* iL3s. *n* = 20 iL3s per genotype. **E** Violin plot depicts the time

taken by control vs. *Sst-cat-2⁻/⁻* iL3s to first puncture the skin. *n* = 21 iL3s per genotype. The dotted line at y = 5 indicates the time at which the assay ended; the dots above this line indicate animals that failed to puncture the skin. **F** Violin plot shows the percentage of pushes or punctures that were followed by backward locomotion that lasted at least 1 s for control vs. *Sst-cat-2⁻/⁻* iL3s. *n* = 21 iL3s per genotype. **G** Violin plot depicts the percentage of penetration attempts, as defined by instances where the worm punctured and partially entered the skin, that were aborted, for control vs. *Sst-cat-2⁻/⁻* iL3s. *n* = 21 control and 11 *Sst-cat-2⁻/⁻* iL3s. For (**C–G**), dots depict individual worms, dashed lines indicate medians, and dotted lines indicate interquartile ranges. Data in (**B–G**) are from 3 independent replicates. A two-tailed Fisher's exact test was used in (**B**) and two-tailed Mann–Whitney tests were used in (**C–G**). Source data are provided in the Source Data file.

less time than wild-type iL3s (Figs. 6C, D and 9C, D, Supplementary Figs. S10C, D, S14C, D). Continued pushing, coupled with the secretion of metalloproteases that help break down the skin[15], leads to skin penetration. Our results also suggest that dopamine signaling suppresses mechanosensory behaviors that would otherwise prevent skin penetration. *Sst-cat-2⁻/⁻* iL3s often reversed after a push or puncture, whereas wild-type iL3s did not (Fig. 6F); these reversals likely prevented the forward locomotion required for skin penetration. The increased tendency of *Sst-cat-2⁻/⁻* iL3s to reverse after pushes and punctures might reflect a hypersensitivity of the mutants to nose touch, as both pushes and punctures are always preceded by head-on collisions with the skin surface. This model is consistent with studies in *C. elegans* that have shown that dopamine signaling modulates the sensitivity to stimuli that cause reversal behavior[28,51,52]. *Sst*-TRP-4 may mediate the effect of dopamine signaling on nose-touch sensitivity, as *Sst-trp-4* mutants also reverse frequently after a push or puncture (Fig. 9F). The dopamine signaling pathway could act constitutively or in a context-dependent manner (*e.g.*, when the worm is on skin) to modulate nose-touch sensitivity. Dopamine might also be required for adjustments in gait as worms transition from crawling on the skin surface to burrowing within the skin tissue; an inability to make these adjustments could lead to the relatively high rate of aborted penetration attempts observed in *Sst-cat-2⁻/⁻* iL3s (Fig. 6G). Consistent with this idea, dopamine has been shown to modulate gait transitions in *C. elegans* in response to changes in the local environment[53,54].

We also asked whether there were differences in skin-penetration behavior on host vs. non-host skin. We found that *S. stercoralis* was equally able to penetrate host human skin and non-host rat skin (Fig. 2D). In contrast, more *S. ratti* iL3s penetrated host rat skin than non-host human skin (Fig. 2H). Both *S. stercoralis* and *S. ratti* iL3s

pushed and punctured the skin of their hosts more frequently than non-host skin (Fig. 2B, F). However, *S. stercoralis* iL3s pushed down on host human skin earlier than non-host rat skin, whereas *S. ratti* iL3s pushed down on both skin types at roughly the same time (Fig. 2C, G). Based on these findings, we suggest that skin-penetration behavior and ability are modulated by at least two categories of factors: (1) worm-intrinsic factors, including the ability to detect host-specific cues such as skin odorants[55–57]; the ability to detect skin-surface openings; and the ability to burrow into skin tissue, which in turn might depend on the body mechanics of the worm and the amount and type of secreted metalloproteases[15], and (2) skin-intrinsic factors, including the presence of host-specific cues, the presence of skin-surface structures that might trigger a push, and the stiffness of the skin.

The skin used in the ex vivo skin-penetration assays might not offer the full complexity of stimuli that is experienced by iL3s on live skin. For instance, the temperature of live skin fluctuates between 30 and 35 °C[58], whereas our assays were conducted at room temperature (23 °C). Heat strongly stimulates activity of *S. stercoralis* iL3s[59], and thus it is possible that iL3s might more quickly penetrate live skin than skin explants. We also manually pluck the fur from the surface of rat skin and exfoliate the surface of human skin prior to performing these assays. These manipulations might alter the distribution of sebum and other lipids that are present in the hair follicle and on the skin surface[60], which might affect the dynamics of skin-penetration behavior. Indeed, it has been shown that skin lipids modulate penetration by the hookworm *Necator americanus*[61]. Mechanical features of the skin such as elasticity and permeability may also change upon separation from the body and freeze-thawing. Nonetheless, the ex vivo skin-penetration assay offers a powerful starting point to examine the interactions between skin-penetrating nematodes and the skin barrier.

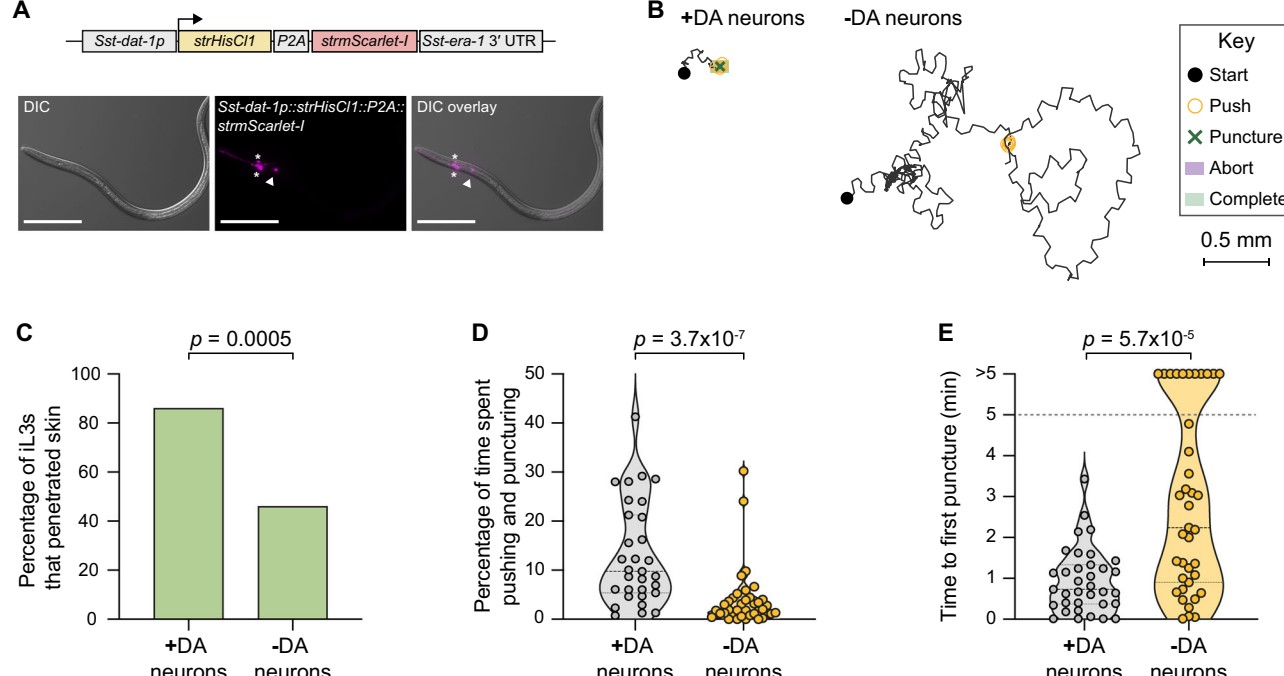

**Fig. 7 | Dopaminergic neurons are required for skin penetration. A** Schematic of the transgene used for silencing *S. stercoralis* dopaminergic (DA) neurons. The histamine-gated chloride channel HisCl1 was expressed in the DA neurons using the *Sst-dat-1* promoter. Exposure of transgenic iL3s to exogenous histamine partially silences the DA neurons (-DA neurons) relative to the vehicle control (+DA neurons). *Sst-dat-1p*, *S. stercoralis dat-1* promoter; *strHisCl1*, *Strongyloides*-codon-optimized *HisCl1*; *P2A*, sequence encoding the self-cleaving P2A peptide; *strmScarlet-I*, *Strongyloides*-codon-optimized *mScarlet-I*; *Sst-era-1* 3' UTR, 3' UTR of *Sst-era-1*. The *Sst-dat-1* promoter is described in Fig. 5B. Montage shows a transgenic iL3 expressing the *Sst-dat-1p::strHisCl1::P2A::strmScarlet-I* construct in the putative *Sst-*CEP (asterisks) and *Sst-*ADE (arrowhead) neurons. Expression in *Sst-*PDE neurons was rarely observed; thus, these neurons may not be silenced. Ventral is up; head is left. Scale bar = 100 μm. **B** Silencing the DA neurons inhibits skin penetration. Behaviors of a representative mock-treated iL3 (+DA neurons) that punctured and completed penetration and a representative histamine-treated iL3 (-DA neurons)

that neither punctured nor completed penetration are shown. Key shows behavioral motifs. **C** Bar graph shows the percentage of mock-treated (+DA neurons) and histamine-treated (-DA neurons) iL3s that penetrated. *n* = 38 +DA-neuron and 39 -DA-neuron iL3s. There are no error bars because the graph shows the percentage of iL3s that completed penetration out of the total tested. **D** Violin plot depicts the percentage of time on skin spent engaging in pushes or punctures. *n* = 31 +DA-neuron and 38 -DA-neuron iL3s. **E** Violin plot depicts the time taken to first puncture skin. *n* = 35 +DA-neuron and 39 -DA-neuron iL3s. The dotted line at y = 5 indicates the time the assay ended; dots above this line indicate animals that failed to puncture. For (**D**, **E**), dots depict individual worms, dashed lines indicate medians, and dotted lines indicate interquartile ranges. Data in (**C–E**) are from 6 independent replicates. A two-tailed Fisher's exact test was used in (**C**), and two-tailed Mann–Whitney tests were used in (**D**, **E**). DiI-stained iL3s and rat skin were used in these experiments. Source data are provided in the Source Data file.

Furthermore, ex vivo skin-penetration assays might be an efficient first pass to screen for drugs that inhibit skin penetration; promising drugs could then be further tested for efficacy in vivo by performing skin-penetration assays on live animals, as we have shown (Fig. 3E, F).

In summary, our results illuminate the complex sequence of behaviors executed by skin-penetrating nematodes on skin and highlight the key role of dopamine signaling in driving these behaviors. Our study also demonstrates that exposing skin-penetrating nematodes to pharmacological inhibitors of dopamine signaling can block skin invasion. While topical insect repellents are widely employed to prevent the spread of insect-transmitted diseases, the possibility of developing topical repellents for parasitic nematodes has not been explored. Our results illustrate the potential for topically applied compounds that block TRP-4 or other nematode-specific components of the dopaminergic pathway to be developed into novel anti-nematode prophylactics.

## Methods
### Ethics statement
All animal protocols and procedures were approved by the UCLA Office of Animal Research Oversight (Protocol ARC-2011-060). The protocol follows the guidelines set by the AAALAC and the *Guide for the Care and Use of Laboratory Animals*. Human skin samples were collected following approval by the University of California

Institutional Review Board (Protocol 22-000400), with signed written informed consent obtained in accordance with the Declaration of Helsinki principles.

### Strains
The following *S. stercoralis* strains were used in this study: UPD, EAH435 *bruIs4[Sst-act-2p::strmScarlet-I::Sst-era-1 3' UTR][19]*, EAH477 *Sst-cat-2(bru3[Sst-act–2p::strmScarlet-I])* II / *Sst-cat-2(bru4[Sst-act-2p::strGFP])* II, and EAH489. EAH489 is a mixed population consisting of worms of the following genotypes: *Sst-trp-4(bru5[Sst-act-2p::strmScarlet-I])* II / +, *Sst-trp-4(bru6[Sst-act–2p::strElectra2::P2A::strElectra2])* II / +, *Sst-trp-4(bru5[Sst-act-2p::strmScarlet-I])*II / *Sst-trp-4(bru5[Sst-act-2p::strmScarlet-I])* II, *Sst-trp-4(bru6[Sst-act–2p::strElectra2::P2A::strElectra2])* II / *Sst-trp-4(bru6[Sst-act–2p::strElectra2::P2A::strElectra2])* II, *Sst-trp-4(bru5[Sst-act-2p::strmScarlet-I])* II / *Sst-trp-4(bru6[Sst-act–2p::strElectra2::P2A::strElectra2])* II, or + / +. The EAH477 strain is composed of worms that express both mScarlet-I and GFP, mScarlet-I only, and GFP only. The dual-colored worms have the *strmScarlet-I* transgene inserted in one allele of *Sst-cat-2* and the *strGFP* transgene inserted in the other allele; the single-colored worms have the same transgene inserted into both alleles of *Sst-cat-2*. In all cases, the transgenes are inserted into the CRISPR site 5'-GTATCTAATTTTGC-TAATGG-3' in *Sst-cat-2*. EAH489 is composed of worms that express both mScarlet-I and Electra2, worms that express mScarlet-I only, worms that express Electra2 only, and non-transgenic worms; a very small subset

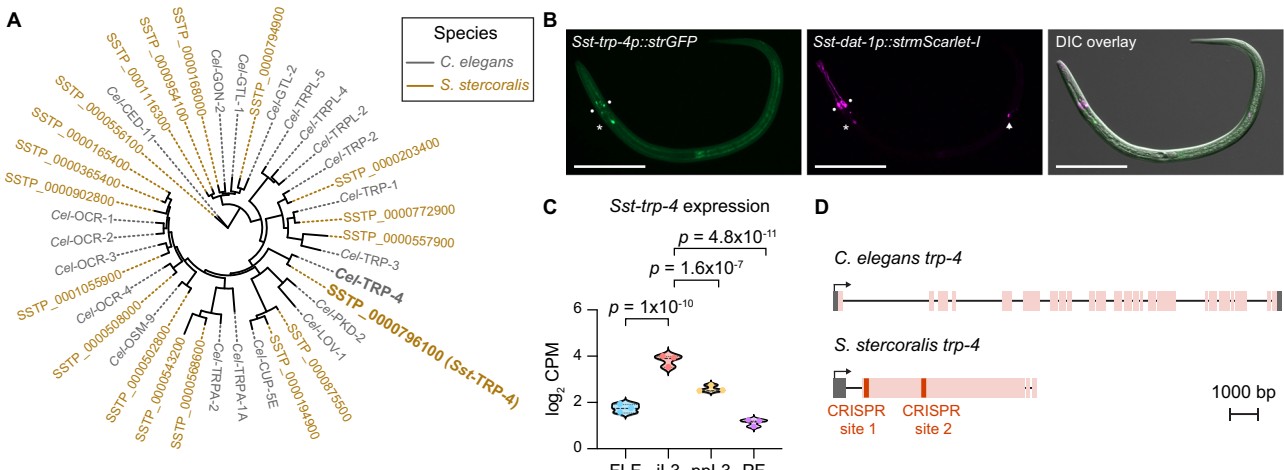

**Fig. 8 | Identification of *S. stercoralis trp-4*. A** Phylogenetic analysis shows the closest homologs of *C. elegans* TRP-4 (gray) in *S. stercoralis* (brown). The tree has all known TRP family members in *C. elegans*[72–74] and the predicted homologs in *S. stercoralis*. **B** Co-expression of *Sst-trp-4* and *Sst-dat-1* in the putative dopaminergic (DA) neurons of *S. stercoralis*. Montage shows expression of the *Sst-trp-4* transcriptional reporter in green, expression of the *Sst-dat-1* transcriptional reporter in magenta, and co-expression of the two reporters and the relative positions of these neurons along the body of the iL3 in the DIC overlay. The *Sst-trp-4* transcriptional reporter comprises a 2320 bp region upstream of the *Sst-trp-4* start codon fused with *strGFP*. The *Sst-dat-1* transcriptional reporter is described in Fig. 5B. The circles, asterisk, and arrow label the putative *Sst*-CEP, *Sst*-ADE, and *Sst*-PDE neurons, respectively; we never observed expression of the *Sst-trp-4* reporter in *Sst*-PDE. Dorsal is up; head is left. Scale bar = 100 μm. **C** Violin plot shows expression levels of

*Sst-trp-4*, in $\log_2$ counts per million (CPM), at the indicated life stages based on published RNA-seq datasets[42,43,68]. Statistical tests for differential gene expression analysis were performed using the limma R package as previously described[42], and *p*-values were adjusted for multiple comparisons using the Benjamini-Hochberg method[42]. FLF free-living female, iL3 infective third-stage larva, ppL3 post-parasitic third-stage larva, PF parasitic female. Each dot indicates an independent replicate. **D** Schematic of the *trp-4* genes of *C. elegans* and *S. stercoralis*. Exons, introns, and UTRs are depicted as pink boxes, black lines, and gray boxes, respectively. Transcriptional start sites are indicated by arrows. *Sst-trp-4* has two CRISPR/Cas9 target sites in the first exon (red); we used two distinct single guide RNAs (sgRNAs), one targeting each CRISPR site, to inactivate *Sst-trp-4*. Drawings are to scale and scale bar = 1000 bp. Source data are provided in the Source Data file.

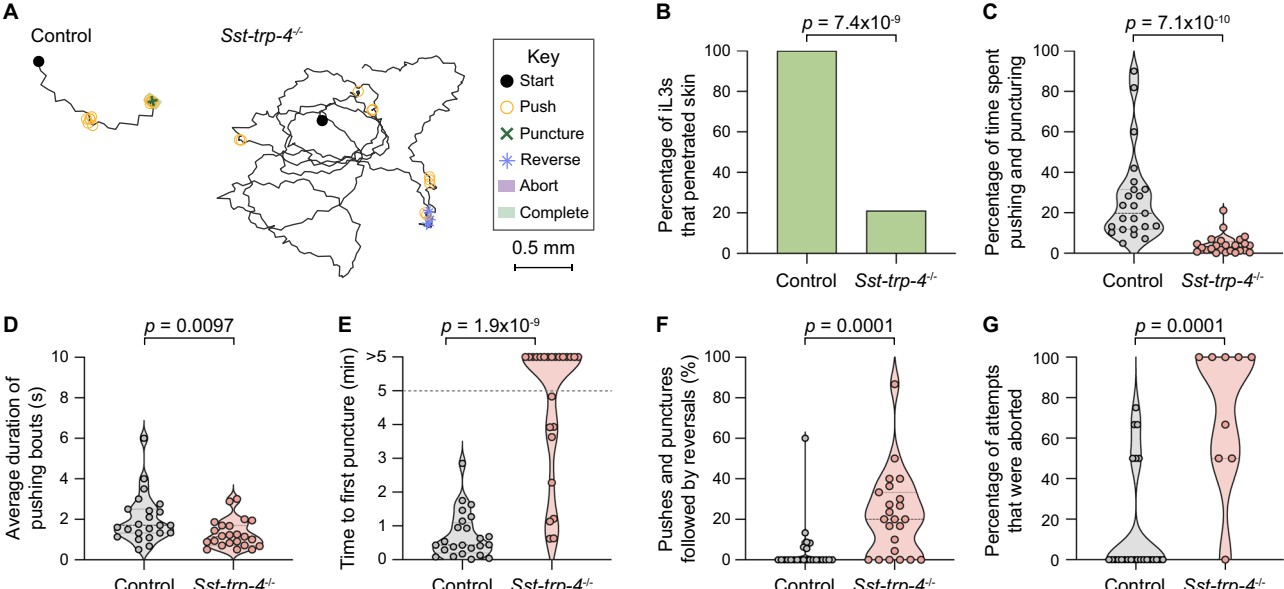

**Fig. 9 | *Sst*-TRP-4 is required for skin penetration. A** *Sst-trp-4* mutants have reduced skin-penetration behavior on rat skin. Tracks show a representative control iL3 that punctured and completed penetration and a representative *Sst-trp-4*[−/−] iL3 that neither punctured nor completed penetration. Key details behavioral motifs. **B** Bar graph shows the percentage of control and *Sst-trp-4*[−/−] iL3s that penetrated. *n* = 24 iL3s per genotype. There are no error bars because the graph shows the percentage of iL3s of each genotype that completed penetration out of the total tested. **C** Violin plot depicts the percentage of time on skin spent engaging in pushes or punctures. *n* = 23 control and 24 *Sst-trp-4*[−/−] iL3s. **D** Violin plot depicts the average duration of pushing bouts. *n* = 23 iL3s per genotype. **E** Violin plot depicts the time taken to first puncture the skin. *n* = 24 iL3s per genotype. The

dotted line at y = 5 indicates the time at which the assay ended; dots above this line indicate animals that failed to puncture. **F** Violin plot shows the percentage of pushes or punctures that were followed by backward locomotion that lasted at least 1 s. *n* = 24 control and 23 *Sst-trp-4*[−/−] iL3s. **G** Violin plot depicts the percentage of penetration attempts, as defined by instances where the worm punctured and partially entered the skin, that were aborted. *n* = 24 control and 9 *Sst-trp-4*[−/−] iL3s. For (**C**–**G**), dots depict individual worms, dashed lines indicate medians, and dotted lines indicate interquartile ranges. Data in (**B**–**G**) are from 3 independent replicates. A two-tailed Fisher's exact test was used in (**B**), and two-tailed Mann–Whitney tests were used in (**C**–**G**). Source data are provided in the Source Data file.

(<0.01%) of the worms in the EAH489 strain co-express both mScarlet-I and Electra2. The dual-colored worms have the *strmScarlet-I* transgene inserted in one allele of *Sst-trp-4* and the *strElectra2::P2A::strElectra2* transgene inserted in the other allele and are homozygous knockouts; the single-colored worms either have the same transgene inserted into both alleles of *Sst-trp-4* and are homozygous knockouts, or they are heterozygotes with a transgene inserted into one allele of *Sst-trp-4* and a wild-type copy of the other allele of *Sst-trp-4*. EAH489 is maintained as a mixed population due to difficulty in establishing a host infection with *Sst-trp-4* homozygous knockouts. To obtain dual-colored homozygous knockout iL3s for assays, crosses were set up between mScarlet-I-positive free-living females and Electra2-positive free-living males, and vice versa. In all cases, the transgenes are inserted between the CRISPR sites 5′-GGCTCTTCAAGTAAACCAGG-3′ and 5′-GCTGATGTTCATTTACATGG-3′ in *Sst-trp-4*. Additionally, experiments wherein dual-colored iL3s were obtained by setting up crosses incorporated, as wild-type controls, wild-type iL3s that were also obtained by setting up crosses between wild-type free-living adults. For *S. stercoralis* skin-penetration experiments that did not involve DiI staining or gene disruptions, the EAH435 strain was used. For skin-penetration experiments that involved DiI staining, the UPD strain was used. For experiments involving knockout *S. stercoralis* iL3s, the EAH435 strain was used as the wild-type control. The following *S. ratti* strains were used in this study: ED231 (wild type) and EAH414 *bruIs3[Sst-act−2p::strmScarlet-I::Sst-era-1* 3′ UTR][19]. For *S. ratti* skin-penetration experiments that did not involve DiI staining, the EAH414 strain was used. For skin-penetration experiments that involved DiI staining, the ED231 strain was used. The *A. ceylanicum* Indian strain (US National Parasite Collection Number 102954) was also used in this study. Genetically modified nematodes used in this study are available upon request from the corresponding author; no MTA is necessary.

## Maintenance of *Strongyloides stercoralis*
*S. stercoralis* was serially passaged in Mongolian gerbils (*Meriones unguiculatus* Strain 243, Charles River Laboratories). Outside of the gerbil, *S. stercoralis* was maintained on fecal-charcoal plates, as previously described[56]. Gerbil infections were done by first collecting *S. stercoralis* iL3s from fecal-charcoal plates using a Baermann apparatus[62] and then washing the iL3s 5 times in sterile 1X PBS. After the last wash, the worms were resuspended in 1X PBS at a concentration of ~10 worms/μL. Each gerbil was anesthetized using isoflurane and then inoculated by subcutaneous injection of 200 μL of the worm/PBS suspension, resulting in an infective dose of ~2000 iL3s/gerbil; 8–12 gerbils at a time were used for strain maintenance. Feces were collected from days 14 to 44 post-inoculation by housing gerbils overnight on wire racks, over damp cardboard lining (Shepherd Techboard, 8 × 16.5 inches, Newco, 999589), in cages. Each morning, feces were collected from the cardboard and then mixed with dH₂O and autoclaved charcoal granules (Bone char, 4 lb pail, 10 × 28 mesh, Ebonex). Fecal-charcoal mixtures were packed, on top of damp Whatman paper, into 10 cm Petri plates (VWR, 82050-918) and placed in plastic boxes lined with damp paper towels. These boxes were either placed directly in a 23 °C incubator or kept at 20 °C for two days and then moved to a 23 °C incubator. *S. stercoralis* free-living females for microinjection were collected from fecal-charcoal plates kept either at 25 °C for one day or 20 °C for two days.

## Maintenance of *Strongyloides ratti*
*S. ratti* was serially passaged in outbred Sprague-Dawley rats (*Rattus norvegicus* Hsd:Sprague Dawley® SD®, Inotiv). Outside of the rat, *S. ratti* was maintained on fecal-charcoal plates. Rat infections were done by first collecting *S. ratti* iL3s from fecal-charcoal plates using a Baermann apparatus[62] and then washing the iL3s 5 times in sterile 1X PBS. Each rat was anesthetized with isoflurane and infected, via subcutaneous injection, with ~700 iL3s in 200 μL of sterile PBS; 2–4 rats at a time were

used for strain maintenance. Feces were collected as described above for *S. stercoralis*, except that collections were done from days 7 to 21 post-inoculation. Fecal-charcoal mixtures were made and maintained as described above for *S. stercoralis*.

## Maintenance of *Ancylostoma ceylanicum*
*A. ceylanicum* was serially passaged in male Golden Syrian hamsters (*Mesocricetus auratus* HsdHan®:AURA, Inotiv). Outside of hamsters, *A. ceylanicum* was maintained on fecal-charcoal plates. Each hamster was infected by oral gavage with 60–100 iL3s resuspended in 100 μL of sterile 1X PBS. Between 2 and 8 hamsters at a time were used for strain maintenance. Feces were collected and fecal-charcoal mixtures were made and maintained as described above for *S. stercoralis*.

## Phylogenetic analyses
We identified the putative *Strongyloides* homologs of *Cel*-DAT-1 using a previously described approach[40]. Briefly, the protein sequence of *Cel*-DAT-1 was retrieved from WormBase WS292 and all similar genes in the *S. stercoralis* and *S. ratti* genomes in WormBase Parasite WBPS18 were identified by performing TBLASTN searches. The accuracy of the gene models for the hits in the *S. stercoralis* and *S. ratti* genomes was checked as described[40,63]; any revisions to gene models were done manually in Geneious Prime 2022.2. The protein sequences of each of these genes were then used in reciprocal TBLASTN searches against WS292. This approach identified all members of the sodium neurotransmitter symporter family (SNF) of proteins in *C. elegans*, including *Cel*-DAT-1. A MUSCLE alignment of all the protein sequences retrieved from the three nematode genomes was performed in Geneious Prime 2022.2. The alignment was fed into IQ-TREE (version 1.6.12), which used a VT + I + G4 substitution model to generate a phylogenetic tree[64,65]; the tree was then visualized using the interactive Tree of Life (iTOL)[66]. This phylogenetic tree is shown in Supplementary Fig. S6A. Using this approach, we identified SSTP_0000953300 and SRAE_2000329000 as the *S. stercoralis* and *S. ratti* homologs of *Cel*-DAT-1, respectively. Additionally, the monoamine transporter domain was identified in both *Cel*-DAT-1 and *Sst*-DAT-1 by performing CD-Search[67] with corresponding protein sequences in the Conserved Domain Database.

The same approach was used to identify the *Strongyloides* homologs of *Cel*-CAT-2. Here, the isoform A of *Cel*-CAT-2 was used in TBLASTN searches because it is the longest isoform. We identified SSTP_0000150600 and SRAE_2000149200 as the *S. stercoralis* and *S. ratti* homologs of *Cel*-CAT−2, respectively. The aromatic amino acid hydroxylase domain in *Cel*-CAT-2 and *Sst*-CAT-2 (SSTP_0000150600) were identified using the CD-Search tool in the Conserved Domain Database[67]. Additionally, the version of *Sst*-CAT-2 annotated in WBPS18 was missing an upstream exon based on publicly available RNA-seq datasets[43,68]. This upstream exon was added manually to the front of the SSTP_0000150600 gene using Geneious Prime 2022.2. The phylogenetic tree showing known members of the biopterin-dependent aromatic amino acid hydroxylases in *C. elegans*, of which CAT-2 is a member, and the predicted homologs in *S. stercoralis* and *S. ratti* is shown in Fig. 5A.

A similar approach was used to identify the *S. stercoralis* homolog of *Cel*-rab-3[69–71]. The longer isoform of *Cel*-RAB-3, isoform A, was used in TBLASTN searches against the *S. stercoralis* genome in WormBase Parasite WBPS18. The protein sequence of the top hit was SSTP_0000451500. We manually removed the SSTP_0000451500 exon 1, as annotated in WBPS18, because the existence of this exon was not supported by RNA-Seq data[43,68]. We then performed a reciprocal TBLASTN search with the corrected SSTP_0000451500 sequence against WS295. This reciprocal BLAST analysis produced *Cel-rab-3* as the top hit, further indicating that SSTP_0000451500 is the *S. stercoralis* homolog of *Cel-rab-3*.

This approach was also used to identify the *Strongyloides* homologs of *C. elegans* TRP channels[72–74]. The protein sequences of *C. elegans* TRP channels were retrieved from WS292 and used in TBLASTN searches. We identified SSTP_0000796100 and SRAE_200231000 as the *S. stercoralis* and *S. ratti* homologs of *Cel*-TRP-4, respectively. Additionally, the version of SSTP_0000796100 annotated in WBPS18 was missing a 5′ UTR based on publicly available RNA-seq datasets[43,68]. This 5′ UTR was added manually to the front of the SSTP_0000796100 gene using Geneious Prime 2022.2. While building the tree of the *C. elegans* and *S. stercoralis* TRP channels, we found a putative novel, longer isoform of the gene SSTP_0000543200, in which the third intron was not spliced, based on RNA-seq data[43,68]. Using the same RNA-seq datasets[43,68], we also found a putative upstream exon in the gene SSTP_0000194900. We manually revised these gene models in Geneious Prime 2022.2.

To build the phylogenetic tree with the TRP channels in *C. elegans*, *S. stercoralis*, *S. ratti*, humans, and mice, we first retrieved the protein sequences of known *H. sapiens* and *M. musculus* TRP family members[75] from UniProtKB (accession numbers are listed in Supplementary Table S2). A MUSCLE alignment of all the protein sequences was done in Geneious Prime 2022.2. As described above, the alignment was then fed into IQ-TREE (version 1.6.12) to generate a phylogenetic tree[64,65], using the same substitution model as above. The tree was visualized using the interactive Tree of Life (iTOL)[66].

We used the same approach as above to identify the *Ancylostoma ceylanicum* homologs of *C. elegans* TRP channels[72–74]. The protein sequences of *C. elegans* TRP channels were retrieved from WS292 and used in TBLASTN searches against the *A. ceylanicum* genome (PRJNA23179) in WormBase Parasite WBPS19. The protein sequences were aligned, and the phylogenetic tree was built and visualized as described above[64–66].

## Molecular biology

The promoter of *Sst-dat*-1 in the plasmid pAGR02 (*Sst-dat-1p::strHisCl1::P2A::strmScarlet-I::Sst-era-1* 3′ UTR) was made by PCR-amplifying the 2579 bp region immediately upstream of the start codon of *Sst-dat-1* (SSTP_0000953300, WBPS18) using the primers RP15 and RP16. The PCR product was then cloned upstream of the *strHisCl1::P2A::strmScarlet-I::Sst-era-1* 3′ UTR cassette, which is contained in pMLC207, using the restriction enzymes HindIII and AgeI. To generate pAGR04 (*Sst-dat-1p::strmScarlet-I::Sst-era-1* 3′ UTR), the *Sst-dat-1* promoter fragment from pAGR02 was excised using the restriction enzymes HindIII and AgeI and then cloned upstream of the *strmScarlet-I::Sst-era-1* 3′ UTR cassette, which is contained in the plasmid pMLC201, using these same restriction enzymes.

The promoter of *Sst-cat-2* in the plasmid pRP19 (*Sst-cat−2p::strGFP::Sst-era-1* 3′ UTR), corresponding to the region 1639 bp upstream of the new start codon of *Sst-cat-2* (SSTP_0000150600, WBPS18) was synthesized by GenScript. GenScript then cloned this promoter fragment upstream of the *strGFP::Sst-era-1* 3′ UTR cassette, which is contained in pMLC200, using the restriction enzymes HindIII and AgeI to generate pRP19.

*Cel-rab-3* is expressed pan-neuronally in *C. elegans*[69–71], and thus we used its predicted homolog in *S. stercoralis* as a general neuronal marker. The promoter of *Sst-rab-3* in the plasmid pRP57 (*Sst-rab-3::strElectra2::P2A:strElectra2*), corresponding to the region 3000 bp upstream of the new start codon of *Sst-rab-3* (SSTP_0000451500), was synthesized by GenScript. GenScript then cloned this promoter fragment upstream of the *strGFP::Sst-era-1* 3′ UTR cassette, which is contained in pMLC200, using the restriction enzymes MluI and SalI to generate pMLC183. The *strGFP* was replaced with a *strElectra2::P2A::strElectra2* cassette using the restriction enzymes AgeI and AvrII to generate pRP57.

The promoter of *Sst-trp-4* in the plasmid pRP37 (*Sst-trp-4p::strGFP::Sst-era-1* 3′ UTR), corresponding to the region 2320 bp upstream of the start codon of *Sst-trp-4* (SSTP_0000796100, WBPS18), was synthesized by GenScript. GenScript then cloned this promoter fragment upstream of the *strGFP::Sst-era-1* 3′ UTR cassette, which is contained in pMLC200, using the restriction enzymes EcoRV and AgeI to generate pRP37.

To generate pRP23 (*Sra-U6p::Sst-cat-2 sgRNA::Sra-U6* 3′ UTR), which is the plasmid that contains the CRISPR single guide RNA (sgRNA) for *Sst-cat-2*, we first used the Find CRISPR sites tool in Geneious Prime 2022.2 to search for guide RNA sites that matched the consensus sequence 5′-GN(17)GG-3′[76]. We found one such site in the second exon of *Sst-cat-2*, with the sequence 5′-GTATC-TAATTTTGCTAATGG-3′ and a Doench et al.[77] activity score of 0.538. This CRISPR RNA sequence was synthesized and fused with the promoter of the *S. ratti* U6 gene, the sequence of the sgRNA scaffold, and the *S. ratti* U6 3′ UTR by GenScript, as previously described[76]. The plasmid pRP31 is the homology-directed repair (HDR) cassette that was used to insert *Sst-act−2p::strmScarlet-I::Sst-era-1* 3′ UTR into the *Sst-cat-2* gene. To make this plasmid, a 479 bp fragment immediately upstream (5′ homology arm) and a 514 bp fragment immediately downstream (3′ homology arm) of the Cas9 cut site in *Sst-cat-2* were first synthesized by GenScript. The 5′ homology arm was cloned upstream of the *Sst-act-2p::strmScarlet-I::Sst-era-1* 3′ UTR cassette, which is contained in pRP12, using the enzymes HindIII and KpnI to generate pRP30. The 3′ homology arm was cloned downstream of the *Sst-act-2p::strmScarlet-I::Sst-era-1* 3′ UTR cassette in pRP30 using the enzymes EagI and BamHI to generate pRP31. The plasmid pRP44 contains the other HDR cassette, *Sst-act-2p::strGFP::Sst-era-1* 3′ UTR, which was made by replacing *strmScarlet-I* in pRP31 with *strGFP* from pMLC200 using the restriction enzymes AgeI and AvrII. Generation of the plasmid pPV540 (*Sra-eef1-1ap::strCas9::Sst-era-1* 3′ UTR) was previously described[76].

To generate pRP41 (*Sra-U6p::Sst-trp-4 sgRNA1::Sra-U6* 3′ UTR) and pRP42 (*Sra-U6p::Sst-trp-4 sgRNA2::Sra-U6* 3′ UTR), which are the plasmids that contain CRISPR sgRNAs that target sites 1 and 2, respectively, in *Sst-trp-4* (Fig. 8D), we first used the Find CRISPR sites tool in Geneious Prime 2022.2, as described above, to identify CRISPR/Cas9 target sites. We identified site 1 (5′-GGCTCTTCAAG-TAAACCAGG-3′) and site 2 (5′-GCTGATGTTCATTTACATGG-3′), which had Doench et al.[77] activity scores of 0.73 and 0.702, respectively. The CRISPR RNA sequences were synthesized and fused with the promoter of the *S. ratti* U6 gene, the sequence of the sgRNA scaffold, and the *S. ratti* U6 3′ UTR by GenScript, as previously described[76]. The plasmid pRP40 is the HDR cassette that was used to insert *Sst-act-2p::strmScarlet-I::Sst-era-1* 3′ UTR into the *Sst-trp-4* gene. To make this plasmid, a 511 bp fragment immediately upstream (5′ homology arm) and a 555 bp fragment immediately downstream (3′ homology arm) of CRISPR sites 1 and 2, respectively, in *Sst-trp-4* were first synthesized by GenScript. The 5′ homology arm was cloned upstream of the *Sst-act-2p::strmScarlet-I::Sst-era-1* 3′ UTR cassette, which is contained in pRP12, using the enzymes HindIII and KpnI to generate pRP38. The 3′ homology arm was cloned downstream of the *Sst-act-2p::strmScarlet-I::Sst-era-1* 3′ UTR cassette in pRP38 using the enzymes EagI and BamHI to generate pRP40. The plasmid pRP50 contains the other HDR cassette, *Sst-act-2p::strElectra2::P2A::strElectra2::Sst-era-1* 3′ UTR. To make this plasmid, the *Strongyloides* codon-optimized version of *Electra2*[78] was first synthesized in the form of *strElectra2::P2A::strElectra2* by GenScript and inserted into the pRP31 backbone using the restriction enzymes AgeI and AvrII. This generated *Sst-act-2p::strElectra2::P2A::strElectra2::Sst-era-1* 3′ UTR. The 5′ homology arm for inactivation of *Sst-trp-4* was inserted upstream of the reporter cassette in pRP31 using the enzymes HindIII and KpnI to generate pRP49. The 3′ homology arm for inactivation of *Sst-trp-4* was inserted downstream of *Sst-act−2p::strElectra2::P2A::strElectra2::Sst-era-1* 3′ UTR in pRP49, using the restriction enzymes EagI and BamHI, to generate pRP50.

## Single-worm skin-penetration tracking assays with *S. stercoralis* and *S. ratti* on rat skin

For single-worm skin-penetration tracking assays on rat skin (Fig. 1A), skin (from the epidermis to the hypodermis) was retrieved from the dorsal and lateral sides of either male or female Sprague-Dawley rats that were 3–13 months old (see below). Euthanized rats that were used for skin retrieval had been frozen at −80 °C either 0 or 1 times; if they were previously frozen, they were allowed to thaw overnight, at room temperature, before the skin was harvested. After retrieval of the skin, it was sectioned into small pieces that were ~2 cm × 2 cm. The skin was then either frozen at −80 °C or used immediately for skin-penetration assays; all skin used in this paper, except the fresh rat skin samples in Supplementary Fig. S2C, had been frozen at least once and up to two times. Fresh rat skin samples were retrieved from rats that had been euthanized 30 min–2 h prior to the start of the skin-penetration assay. For skin-penetration assays, pieces of skin were first allowed to equilibrate to room temperature and fur was then manually plucked from the surface of the skin. Skin sections were then draped over plastic cell culture inserts (CellCrown) and placed in individual wells of either 12-well or 6-well plates. If a 12-well plate (VWR Scientific, 10062-894) was used, the skin was draped over cell culture inserts made for 24-well plates (Millipore Sigma, Z742381). If a 6-well plate (Corning, 3516) was used, the skin was placed on cell culture inserts made for 12-well plates (Millipore Sigma, Z742383). The wells were filled with 1–3 mL of BU saline[79] prior to placing the skin and insert in the well to ensure moisture retention within the skin. For some assays, the skin was air-dried for 5–15 min before performing assays to remove excess moisture on the skin surface. If the skin was slightly dried for assays, both control and experimental groups were tested on skin that was dried to the same extent.

While the skin was equilibrating to room temperature, *S. stercoralis* or *S. ratti* iL3s were isolated from fecal-charcoal plates using a Baermann apparatus[62], as previously described. The fecal-charcoal plates used for collection from stable lines were 5–10 days old. The plates used for collection of transgenic *S. stercoralis* iL3s from free-living adults that were microinjected were 5–7 days old. If worms were stained with DiI prior to skin-penetration assays, a ~50–100 μL worm pellet from the Baermann apparatus was first resuspended in ~10 mL of BU saline[79] and this worm suspension was then used for dye staining. One mL of the suspension was dyed with 5 μL of DiI (2 mg/mL, Thermo Fisher Scientific, D3911) for 15 min prior to skin-penetration assays; the solvent for DiI was N,N-dimethylformamide (DMF, Thermo Fisher Scientific, 68-12-2) and DiI solutions used for staining were no older than 3 months old. After staining, the worms were washed twice in fresh BU[79] to remove excess DiI and DMF from the surface of the worm. Next, the worms (either DiI-stained iL3s or unstained, transgenic iL3s) were plated onto 10 cm unseeded nematode growth medium (NGM) plates[80]. Individual worms were picked from NGM plates using a paintbrush and allowed to crawl onto the skin surface. Immediately thereafter, the worms on skin were recorded using a Leica M165 FC fluorescence dissection microscope, with the ET-mCherry filter set (Leica, 10450195), and an attached Basler ace (either acA3800-14um or acA5472-17um) camera for either 5 or 10 min or until skin penetration was complete. A series of still images was captured at 4 frames/second (fps) and image acquisition was controlled with the pylon Viewer software (Basler). The field of view was manually adjusted whenever the worm moved out of the field of view of the microscope during the course of the assay. Up to 10 worms were assayed, sequentially, on the same piece of skin. Assays were performed blind to genotype and blinding was lifted after the experiment was over.

Images captured for each worm were opened in Fiji 2.9.0/1.53t[81] using File>Import>Image Sequence. Although images were captured at 4 fps, every alternate image was opened in Fiji as this resolution was sufficient for tracking behavior. All behaviors that are described were

scored manually. Worms normally pushed the skin and then either crawled on the surface or punctured the skin. A bout of pushing was defined as the first frame at which the worm was observed to be pushing on the skin until the frame just prior to the one in which the worm either was crawling or had punctured the skin. A worm was considered to have punctured the skin if it fulfilled one or both of the following conditions: (1) the nose of the worm was detected inside the skin, as indicated by a decrease in fluorescence and blurring of the nose, or (2) the nose had disappeared from the skin surface while the worm moved into the skin. Additionally, a worm that had punctured the skin appeared anchored to the skin surface by its head while the rest of its body moved freely. Worms were considered to be reversing if they moved backwards for 2 or more frames (i.e., 1 s or more). A reversal was considered to be associated with a push or puncture if it occurred within 5 frames (i.e., 2.5 s or less) of the latter behaviors. Consecutive frames of backward movement that lasted for at least 2 frames (i.e., 1 s or more) and were interrupted for fewer than 4 frames were ascribed to the same reversal bout. The proportion of time spent by individual worms pushing or puncturing the skin was only plotted for those worms that had not already initiated penetration by the time of recording.

To generate tracks of worms on skin, worm position was tracked manually in Fiji using the TrackMate plugin[82] and plotted using custom MATLAB software that can be accessed at this URL: https://github.com/BryantLabUW/WormTracker3000. In cases where the field of view was adjusted during the course of the assay, coordinates of the worm in each field of view were tracked independently. Worm tracks in each field of view were plotted separately using the abovementioned MATLAB software. The end of a worm track in the first field of view was then overlaid with the beginning of the track in the subsequent field of view; this was repeated until all the tracks generated for a given worm formed a single, continuous track. For the representative worms shown in the figures, some completed skin penetration while others did not. In both cases, representative worms were defined as those whose time spent pushing and puncturing was close to the median value of the entire cohort.

The images of the iL3 in Fig. 1 and the movie of the same iL3 (Supplementary Movie S1) were generated by recording an EAH435 iL3 on the surface of rat skin using a Zeiss Axio Zoom V16 (Zeiss Plan-NeoFluar Z 1 × /0.25 FWD 56 mm objective), with the 43 HE dsRed filter, and an attached Basler ace (acA5472-17um) camera until skin penetration was complete. As described above, a series of still images was captured at 4 fps and image acquisition was controlled with the pylon Viewer software (Basler).

## Rats used for ex vivo skin-penetration assays

All rats used in ex vivo skin-penetration assays were outbred Sprague-Dawley rats (*Rattus norvegicus* Hsd:Sprague Dawley® SD®, Inotiv). Three 10-month-old male rats were used for skin-penetration assays in Fig. 2A–D. An 8-month-old male rat was used for Fig. 2E–H. One 2-month-old female rat, one 3-month-old female rat, and one 12-month-old male rat were used for Fig. 3A–D. A 9-month-old male rat was used for Fig. 4. Two 11-month-old male rats were used for Fig. 6. One 13-month-old female rat, one 5-month-old male rat, one 9-month-old male rat, one 12-month-old male rat, and one 13-month-old male rat were used for Fig. 7. Three male rats of unknown ages were used for Fig. 9. One 13-month-old female rat, one 8-month-old male rat, and one 13-month-old male rat were used for Supplementary Fig. S4. Two 8-month-old male rats were used for Supplementary Fig. S5A. Rats of unknown ages and sexes were used for Supplementary Fig. S5B.

## Movement tracking assay for *S. stercoralis* iL3s on agar surfaces

Wild-type and EAH435 *bruIs4[Sst-act-2p::strmScarlet-I:: Sst-era-1 3′ UTR][19]* transgenic *S. stercoralis* iL3s were isolated from 7-8-day-old fecal-charcoal plates on the day of the assay and resuspended in BU saline[79].

DiI staining of wild-type iL3s was performed as described above and DiI-stained iL3s were used for assays within 1 h of staining. Prior to movement tracking, iL3s were plated on a 6 cm NGM plate and allowed to crawl around for at least 5 min. Then, 2–4 iL3s were picked from this plate and placed in a 2 μL droplet of BU saline[79] on a distinct 6 cm NGM plate. This plate was then placed under a Leica M165 FC fluorescence dissection microscope and movement was recorded using a Basler ace (acA3800-14um) camera. Images were acquired at 2 frames/s for a total of 1 min after the worms cleared the droplet of BU saline. The frames corresponding to the last 30 s of the recording were opened in Fiji and the coordinates of the worm in these frames were tracked manually using the TrackMate plugin. The speeds and path lengths were then calculated from these coordinates using custom MATLAB software (https://github.com/BryantLabUW/WormTracker3000).

## Human skin sample collection

Human skin was obtained either commercially from the forearm of a cadaver donor (Accio Biobank) or from live donors through plastic surgery. The cadaver donor was a 65-year-old male; the cause of death was acute myeloid leukemia. The skin was retrieved from the donor within 8 h after death, frozen, and shipped on dry ice. In the case of live donors, skin was obtained from adult patients (30–50 years old, either male or female) who were undergoing elective plastic surgery at the UCLA Dermatology Clinic. Fresh skin samples were suspended in 1X PBS immediately after retrieval from the donor. In all cases, the skin, which was normal in appearance, was sectioned (after thawing, in the case of the cadaver donors) and frozen at −80 °C.

## Single-worm skin-penetration tracking assays with *S. stercoralis* and *S. ratti* on human skin

For single-worm skin-penetration tracking assays on human skin (Supplementary Fig. S3), human skin samples were thawed slowly overnight at 4 °C and then allowed to equilibrate to room temperature for 1–2 h before performing skin-penetration assays. On the day of the skin-penetration assay, transgenic *S. stercoralis* or *S. ratti* iL3s were isolated using a Baermann apparatus[62], as previously described. We then prepared the thawed skin samples. Infective larvae usually come in contact with human hosts when humans are walking barefoot through contaminated soil[7]. To mimic the thinner skin of the dorsum of the foot[83,84], which is thought to be the major region of entry for skin-penetrating nematodes[6], as well as any abrasions that might be caused by walking barefoot, we first exfoliated the human skin samples using an Amope Pedi Perfect electronic foot file that was fitted with the head roller attachment that had ultra coarse grains (#4) for 5–15 s. The exfoliated skin was then placed between cotton pads that were pre-moistened with 1X PBS in a 6 cm Petri plate (Tritech Research, T3315) and used for assays within 1 h of exfoliation. The cotton pad from the top of the skin was removed and the skin surface was lightly blotted with a tissue wipe (VWR International, 82003-820) just prior to the start of the assay. Immediately after blotting, individual worms were placed on the skin surface and time-lapse images were acquired for 10 min afterward or until penetration was complete, as described above for the rat skin assays. Up to 4 worms were assayed on the same piece of skin consecutively, and the skin surface was re-moistened with 1X PBS and blotted lightly between each worm. Assays were performed blind to genotype and blinding was lifted after the experiment was over.

## Skin-penetration assays with *S. ratti* iL3s on live rats

Live Sprague-Dawley rats that were 8–10 months old (see below) were used for in vivo skin-penetration assays. The live rats were anesthetized by exposure to isoflurane (Sigma-Aldrich, 26675-46-7). The entire rat was first placed in an isoflurane exposure chamber (Somni Scientific) for ~4–5 min. Once the animal appeared anesthetized, it was transferred onto a small animal heating pad (K&H, HM10) and

isoflurane was continuously provided, via a customized nose cone (Patterson Scientific, 78909683) that enclosed just the nose and mouth of the animal, for the remainder of the assay. Rats were used for assays for up to 100 min following initial isoflurane exposure; skin surface temperature was monitored during this entire time period and fluctuated between 30 and 35 °C.

A 2 cm × 2 cm section of fur-free skin was prepared by manually plucking the fur from the skin surface. The rats often had a brown undercoat beneath the white fur, which was removed by scrubbing with a cotton swab soaked in BU saline[79]. The fur-free section of skin was then moisturized with 5 μL of paraffin oil (Supelco, 8012-95-1). Next, individual transgenic *S. ratti* iL3s, soaked overnight in either DMSO or haloperidol (see below), were placed on the skin surface and visually tracked for punctures and penetration using a Leica S6E stereomicroscope fitted with an adapter for red fluorescence visualization (NIGHTSEA, SFA-GR); illumination was provided using two fluorescence excitation light heads (NIGHTSEA, SFA-GR). The worms were observed until one of the three conditions were met: (1) they penetrated the skin; (2) they did not penetrate the skin but stopped moving on the skin surface; (3) they did not penetrate the skin but were moving on the skin for 5 min. Most worms that did not puncture or penetrate the skin moved for at least 1.75 min on the skin surface. Up to 4 worms were sequentially assayed on the same skin section and up to three skin sections were sequentially prepared on the same rat. Data from at least two rats were collected on each experimental day. Both male and female rats were tested.

## Rats used for in vivo skin-penetration assays

All rats used for in vivo skin-penetration assays were outbred Sprague-Dawley rats (*Rattus norvegicus* Hsd:Sprague Dawley® SD®, Inotiv). Two 10-month-old female rats and seven 10-month-old male rats were used for the in vivo skin-penetration assays in Fig. 3E, F.

## Single-worm skin-penetration tracking assays with *A. ceylanicum* on rat skin

Skin-penetration assays with *A. ceylanicum* iL3s were done essentially as described above, with a few differences. *A. ceylanicum* iL3s were typically collected from plates that were 10–14 days old. Worms were stained with DiI, as described above, for 10 min and then immediately spun down at 3000 × g for 1 min. All iL3s were plated on 10 cm NGM plates[80] and allowed to exsheath for 10–15 min. Since DiI strongly stained the sheath, exsheathed worms could be easily identified as the non-fluorescent worms using a Leica M165 FC microscope (ET-mCherry filter set, Leica, 10450195). The exsheathed worms were picked and placed into a watch glass that had 1 mL of BU[79] pre-mixed with 10 μL of DiI. Approximately 10 min later, individual iL3s were pipetted from the watch glass onto a 10 cm NGM plate[80] and then transferred onto skin using a paintbrush. Images were captured, as detailed above, for 5 min or until iL3s had penetrated the skin. Only iL3s that were in the second DiI stain for less than an hour were used for skin-penetration assays.

## Treatment of worms with haloperidol and dopamine

Haloperidol-treatment of *S. stercoralis*, *S. ratti*, and *A. ceylanicum* iL3s was done overnight at room temperature. Stock solutions of either 20 mM or 40 mM haloperidol (Millipore Sigma, 52-86-8) in DMSO (Millipore Sigma, 67-68-5) were made fresh prior to each experiment. For ex vivo skin-penetration assays, *S. stercoralis* iL3s were treated with 1.5 mM haloperidol in BU, whereas *S. ratti* and *A. ceylanicum* iL3s were treated with 160 μM haloperidol in BU. Vehicle-only controls were treated overnight with an equal concentration of DMSO only; the concentrations of DMSO were 0.8%, 3.8% and 0.4% for the *S. ratti*, *S. stercoralis*, and *A. ceylanicum* iL3s, respectively. Skin-penetration assays were then performed as described above. For in vivo skin-penetration assays, *S. ratti* iL3s were treated overnight with either

160 μM or 1 mM haloperidol in BU[79]. Vehicle-only controls were also treated overnight with 2.5% DMSO in BU[79].

Treatment of iL3s with dopamine (DA) was done for 1–2 h. A stock solution of 1 M DA (Millipore Sigma, 62-31-7) in ddH₂O was made fresh on the morning of each assay day and stored in the dark at 4 °C until addition to the worm suspension. To expose the worms to DA, the stock solution of DA was added to the worms (that were being treated either with haloperidol or DMSO) to a final concentration of 10 mM. Skin-penetration assays were performed as described above.

For the haloperidol dose-response experiments in Supplementary Fig. S5, iL3s were treated with the indicated concentrations of halo-peridol or DMSO overnight; the total number of worms that were treated was between 1200 and 2500 in a total volume of 1 mL of liquid. On the day of the assay, 1 mL of the iL3 suspension was stained with 5 μL of DiI (2 mg/mL in DMF) for 15 min, as described above. Individual iL3s were then sequentially placed on rat skin and the time to first puncture was manually recorded using a fluorescence dissection microscope. Worms were used within 1 h of dye staining and up to 5 worms/condition were placed on the same piece of skin. For assays with *S. stercoralis* iL3s, the rat skin was dried for 5 min prior to the start of the assay and used for no more than 2 h after drying; rat skin was not dried for assays with *S. ratti* iL3s.

### Generation of stable mutant lines using CRISPR/Cas9-mediated mutagenesis

The generation of stable mutant lines was done as previously described[41] (Supplementary Fig. S8). To generate the *Sst-cat-2* mutant line, *S. stercoralis* free-living females were collected from fecal-charcoal plates kept at 25 °C for one day or 20 °C for two days and then microinjected[85] with one of the two following injection mixes: mix 1 (for generation of red worms), which had pRP23 (80 ng/μL), pRP31 (80 ng/μL), and pPV540 (50 ng/μL); or mix 2 (for generation of green worms), which had pRP23 (80 ng/μL), pRP44 (80 ng/μL), and pPV540 (50 ng/μL). The plasmid pRP23 supplied the sgRNA targeting *Sst-cat-2*; pRP31 and pRP44 contained HDR cassettes for inserting *Sst-act-2p::strmScarlet-I* and *Sst-act-2p::strGFP*, respectively, at the CRISPR cut sites in *Sst-cat-2*; and pPV540 was the Cas9 expression vector. F₁ iL3s were isolated from fecal-charcoal plates using a Baermann apparatus[62] between 5 and 7 days later. Approximately 100–200 iL3s were plated at a time on 6 cm NGM plates that were seeded with *E. coli* OP50[80] and then screened for full-body mScarlet-I expression or full-body GFP expression using the Leica M165 FC microscope, with the ET-mCherry filter (Leica, 10450195) or the ET GFP filter set (Leica, 10447408), respectively. The transgenic F₁ iL3s were picked, pooled and then activated by incubating them, for ~40–44 h, in Dulbecco's Modified Eagle Medium (DMEM, Gibco, 11995065) at 37 °C and 5% CO₂, as described previously[57]. After incubation, iL3s were collected, washed 3 times in 1X PBS, re-suspended in 200 μL of 1X PBS, and then introduced into a single gerbil by oral gavage. Feces and iL3s were collected from the infected gerbil as described above. The presence of transgenic F₂/F₃ iL3s was confirmed by screening, as described above, and dual-colored worms that expressed both mScarlet-I and GFP were picked and used either for skin-penetration assays, genotyping, or main-tenance of the mutant strain. The occurrence of both mScarlet-I and GFP expression in the same worm implied that both copies of *Sst-cat-2* were inactivated: one copy was inactivated by HDR-based insertion of *Sst-act-2p::strmScarlet-I* and the other by HDR-based insertion of *Sst-act-2p::strGFP*. These dual-colored worms were preferred for assays and strain maintenance because they could be visually confirmed to be homozygous mutants.

A similar approach was used to generate the *Sst-trp-4* mutant line[41]. Free-living females were microinjected[85] with one of the two mixes: mix 1 (for generation of red worms), which had pRP41 (40 ng/μL), pRP42 (40 ng/μL), pRP40 (80 ng/μL), and pPV540 (50 ng/μL); or mix 2 (for generation of blue worms), which had pRP41

(40 ng/μL), pRP42 (40 ng/μL), pRP50 (80 ng/μL), and pPV540 (50 ng/μL). The plasmids pRP41 and pRP42 supplied distinct sgRNAs for targeting *Sst-trp-4*; pRP40 and pRP50 contained HDR cassettes for inserting *Sst-act-2p::strmScarlet-I* and *Sst-act-2p::strElectra2::P2A::str-Electra2*, respectively, between the CRISPR cut sites in *Sst-trp-4*; and pPV540 was the Cas9 expression vector. After 5–7 days, F₁ iL3s were isolated[62] and screened either for full-body mScarlet-I signal or full-body Electra2 signal, using the Leica M165 FC microscope, with the ET-mCherry filter (Leica, 10450195) or the ET BFP2 filter set (Leica, 10450571), respectively. Worms were picked, pooled, and introduced into a gerbil by oral gavage as described above. Due to difficulty in establishing an infection with *Sst-trp-4* homozygous knockout worms, the *Sst-trp-4* line (EAH489) was maintained as a mixed population by selecting red-only, blue-only, and red/blue worms for gerbil infections. Only red/blue homozygous knockout iL3s were used for assays.

### Genotyping worms from CRISPR/Cas9 assays

Worm lysis was done as previously described, wherein individual iL3s were placed in 6 μL of worm lysis buffer (50 mM KCl, 10 mM Tris pH 8, 2.5 mM MgCl₂, 0.45% Nonidet-P40, 0.45% Tween-20, and 0.01% gelatin in ddH₂O) supplemented with ~0.12 μg/μL Proteinase-K (Millipore Sigma, 39450-01-6) and ~1.7% 2-mercaptoethanol (Millipore Sigma, 60-24-2)[76]. For genotyping at the *Sst-cat-2* locus, each lysed worm was used for 3 PCR reactions: 1) a positive control reaction with primers SG78 and SG80 that target the *Sst-act-2* gene and produce a 416 bp amplicon; 2) a reaction with primers RP32 and RP33, which produce a 666 bp band specifically with the wild-type *Sst-cat-2* allele; and 3) a reaction with primers RP30 and RP34, which produce a 690 bp band specifically with the mutant *Sst-cat-2* allele. For genotyping at the *Sst-trp-4* locus, each lysed worm was similarly used for 3 PCR reactions: 1) the same positive control reaction as above that targets the *Sst-act-2* gene; 2) a reaction with primers RP39 and RP46, which produce a 604 bp band specifically with the wild-type *Sst-trp-4* allele; and 3) a reaction with primers RP39 and RP28, which produce a 720 bp band specifically with the mutant *Sst-trp-4* allele. Genotyping primers are listed in Supplementary Table S1. The polymerase PlatTaq (Thermo Fisher Scientific, 10966034) was used, and each reaction had a final volume of 25 μL. The PCR reactions were run on an Eppendorf Mas-tercycler Nexus Gradient (Millipore Sigma, EP6331000025) using the following cycling conditions: initial denaturation 94 °C (2 min); 94 °C (30 s), 53 °C (30 s), 68 °C (1 min) ×35 cycles; final extension 68 °C (5 min); 10 °C (hold). PCR products were run out on a 2% agarose gel and stained with GelGreen (Biotium, 41005); the size of each product was gauged by comparing with a 100 bp ladder (New England Biolabs, N3231). The gels were imaged in a ChemiDoc MP Imaging System (Bio-Rad Laboratories) using an exposure time of 1 s. Images were acquired using Image Lab 5.1 (Bio-Rad Laboratories). The presence of bands in each lane was determined using the Lane and Bands tool with the Band Detection Sensitivity set to 100%.

### Histamine assays

*S. stercoralis* free-living females were collected from fecal-charcoal plates kept at 25 °C for one day or 20 °C for two days and microinjected with pAGR02 at 80 ng/μL using well-established techniques[85]. F₁ iL3s were isolated from fecal-charcoal plates using a Baermann apparatus[62] 5–7 days later. Approximately 100-200 iL3s were plated at a time on 6 cm NGM plates that were seeded with *Escherichia coli* OP50[80] and then screened for mScarlet-I expression in the dopaminergic neurons using a Leica M165 FC microscope with the ET-mCherry filter (Leica, 10450195). Only transgenic iL3s with mScarlet-I signal visible in mul-tiple *Sst*-CEP neurons and the *Sst*-ADE neurons at 20–25× magnification were picked for skin-penetration assays; expression of the construct in the *Sst*-PDE neurons was variable, so these neurons were likely not always silenced in our experiments. The transgenic F₁ iL3s were picked and placed in 1 mL of BU saline[79] and left at 23 °C overnight. The

following day, transgenic worms were split into two batches: one batch was treated with histamine dihydrochloride (stock concentration was 1 M in ddH$_2$O, Millipore Sigma, 56-92-8) diluted to a final concentration of 50 mM in BU[40]; the other batch was treated with an equal volume of ddH$_2$O (the solvent for the histamine stock solution), which was also mixed with BU[40]. Skin-penetration assays were performed, as described above, 4 h later. Assays were performed blind to experimental condition and blinding was lifted after all the worms had been recorded.

## Fluorescence microscopy

Microscopy of worms was performed using previously established methods for fluorescence microscopy of paralyzed nematodes[40]. *S. stercoralis* free-living females were microinjected, as detailed above, and recovered on fecal-charcoal plates. For the images in Figs. 5B and 8B, the corresponding transcriptional reporter constructs were microinjected at 80 ng/µL. For the images in Supplementary Fig. S7, the *Sst-dat-1* and *Sst-rab-3* transcriptional reporters were microinjected at 60 ng/µL and 70 ng/µL, respectively; these microinjection mixes also included pMLC131, which is the expression vector for the hyperactive piggyBac transposase (hyPBase)[19], at 50 ng/µL. We used hyPBase to force genomic integration of the *Sst-dat-1* and *Sst-rab-3* transcriptional reporters, because integrants are more likely to show full, non-mosaic expression of the corresponding transgenes[19,76]. Transgenic F$_1$ iL3s were isolated from these plates after 5–7 days by screening under the Leica M165 FC microscope using either the ET GFP filter set (Leica, 10447408) or the ET mCherry filter set (Leica, 10450195); iL3s were paralyzed with 1% nicotine (Millipore Sigma, 54-11-5) prior to screening. The transgenic iL3s were then exposed to 50 mM levamisole (Millipore Sigma, 16595-80-5) in BU saline[79], mounted on a slide with 5% Noble agar dissolved in BU[79], and covered with a coverslip.

Epifluorescence and DIC images in Figs. 5B, 7A and 8B, Supplementary Figs. S9B and S13B were taken with either a 20× objective (Plan-Apochromat 20 × /0.8 M27; Zeiss) or a 40× oil objective (Plan-Apochromat 40 × /1.4 ∞/0.17 Oil DIC (UV) VIS-IR M27; Zeiss) on an inverted Zeiss AxioObserver microscope equipped with a 38 HE filter set for GFP (BP470/40, FT495, BP 525/50), a 63 HE filter set for mScarlet-I (BP572/25, FT590, BP629/62), a 96 HE filter set for Electra2 (BP 390/40, FT420, BP450/40), and a Hamamatsu ORCA-Flash 4.0 camera; fluorescence illumination was provided by Colibri 7 LEDs (LED-Module 475 nm). All images were captured using Zeiss ZEN 2 (blue edition) software. Images in magenta were pseudo-colored in Fiji 2.9.0/1.53t[81] and image montages were generated in Adobe Photoshop 25.5.0 and Adobe Illustrator 28.4.1. Additionally, images in Figs. 5B, 7A, and 8B are maximum intensity projections of Z-stacks that were generated in Fiji 2.9.0/1.53t[81].

Fluorescence images in Supplementary Fig. S7 were taken with a 40× oil objective (C-Apochromat/1.2 W Korr FCS M27) on an inverted Zeiss LSM 880 confocal laser scanning microscope equipped with a 561 nm laser line for mScarlet-I excitation, a 405 nm laser line for Electra2 excitation, and MA-PMT detectors for signal acquisition. All images were captured using Zeiss Zen 2.1 (black edition) software as a Z-stack in 2.23 µm sections. A maximum intensity projection of the Z-stack and pseudo-coloring were done in Fiji 2.9.0/1.53t[81], and image montages were generated in Adobe Photoshop 26.0.0 and Adobe Illustrator 28.5.

## Statistical analysis and reproducibility

Statistical analyses were performed in Prism 10.0.0. The statistical tests used for each experiment are listed in the figure legends; two-tailed tests were used for all statistical analyses. Non-parametric tests were used when the data were found to be non-parametrically distributed, as determined by tests for normality in Prism. Sample sizes were determined by power analysis using G*Power 3.1.9.6.

The expression profiles of *Sst-cat-2* and *Sst-dat-1* shown in Fig. 5B were also detected in 21 other worms. The expression profile of *Sst-dat-1p::strHisCl1::P2A::strmScarlet-I* shown in Fig. 7A was detected in 4 other worms using the Zeiss AxioObserver microscope. Similar expression profiles were seen at lower magnification, using the Leica M165 FC microscope, in 74 worms. Co-expression of the *Sst-trp-4* and *Sst-dat-1* reporters (Fig. 8B) was detected in two other worms using the Zeiss AxioObserver microscope. This co-expression was also detected, at lower magnification on the Zeiss M165 FC microscope, in 27 worms. Co-expression of *Sst-dat-1p::strmScarlet-I* and *Sst-rab-3p::Electra2::P2A::Electra2* (Supplementary Fig. S7) was detected in 3 additional worms. Co-expression of GFP and mScarlet-I in *Sst-cat-2*$^{-/-}$ dual-colored iL3s (Supplementary Fig. S9B) was imaged on the Zeiss AxioObserver microscope in one additional worm and detected on the Leica M165 FC in 72 additional worms. Similarly, co-expression of mScarlet-I and Electra2 in *Sst-trp-4*$^{-/-}$ iL3s (Supplementary Fig. 13B) was imaged on the Zeiss AxioObserver microscope in three additional worms and detected on the Leica M165 FC in 78 additional worms.

For the genotyping gel shown in Supplementary Fig. S9C, similar banding patterns were detected in 3 additional *Sst-cat-2*$^{-/-}$ dual-colored iL3s and one additional wild-type iL3. For the genotyping gel shown in Supplementary Fig. S13C, similar banding patterns were detected in five additional *Sst-trp-4*$^{-/-}$ dual-colored iL3s and one additional wild-type iL3.

## Reporting summary

Further information on research design is available in the Nature Portfolio Reporting Summary linked to this article.

## Data availability

All data necessary for the conclusions described in this study are included with this article. Source data are provided with this paper. The behavior tracking files and RNA-Seq data for *Sst-cat-2*, *Sst-dat-1*, and *Sst-trp-4* are available from GitHub (https://github.com/HallemLab/Patel_et_al_2025). The raw images of behavior and expression profiles of *Sst-cat-2*, *Sst-dat-1*, *Sst-trp-4*, and *Sst-rab-3* are available on BioImage Archive (https://www.ebi.ac.uk/biostudies/bioimages/studies/S-BIAD1970). Source data are provided with this paper.

## Code availability

Custom code used for skin-penetration tracking assays is available from GitHub (https://github.com/BryantLabUW/WormTracker3000) and is also archived with Zenodo (https://doi.org/10.5281/zenodo.15509108)[86].

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

## Acknowledgements

We thank Navonil Banerjee and Breanna Walsh for thoughtful comments on the manuscript, Tiffany Mao for hand-drawn illustrations, Yanying Dai for help with animal husbandry and animal infections, and the UCLA Broad Stem Cell Research Center Microscopy Core for access to and use of the Zeiss LSM 880. The illustrations shown in Fig. 1 and Supplementary Figs. S1, S3, S8, and S11A were created with BioRender. Gene models in Figs. 5 and 8 were adapted from models made using the Exon-Intron Graphic Maker (http://www.wormweb.org/exonintron). This work was supported by NIH F32AI174816 (R.P.), T32GM145388 and T32AI007323 (G.B.), NIH MARC T34GM008563 (A.G.R.), funds provided by the University of Washington School of Medicine and NIH DP2AI184544 (A.S.B.), NIH R01AR081337 (G.W.A), and NIH R01AI175183 (E.A.H.).

## Author contributions

R.P. and E.A.H. conceived the study. R.P., G.B., M.L.C., and E.A.H. designed experiments. R.P., G.B., M.L.C., and A.G.R. performed experiments. A.S.B. contributed original code. G.W.A. provided human skin samples. R.P. and E.A.H. wrote the manuscript and prepared figures. All authors reviewed and provided feedback on the manuscript.

## Competing interests

The authors declare no competing interests.
