## [Transparent Peer Review file · Nature Communications]

Dopamine signaling drives skin invasion by human-infective nematodes

Corresponding Author: Dr Elissa Hallem

Version 0:

Reviewer comments:

Reviewer #1

(Remarks to the Author)

Patel et al, describe a role for dopamine signalling in skin penetration behaviour in clade V helminths. The authors suggest skin treatment as a preventative approach against skin penetrating nematode. The project is extremely original, well designed and clearly written.

Major comments:

- A major limitation of the study is the use of frozen skin. The authors state doing experiments in both fresh and frozen tissue but do not report the data to evaluate a potential difference. The problem of the dead skin is that the skin permeability will likely be affected and one might wonder whether it then recapitulate a live infection.

- The rate of skin penetration reported is indeed incredibly high (80-100%) as compared to previously reported results of 20 % (*S. ratti* in human skin, doi.org/10.1016/j.exppara.2015.01.005). The bar plot do not have any error bar to assess the variability of the model.

- Could the authors assess the potential of Haloperidol and Dopamine treatment in an in vivo experiment, either by following with live imaging the entrance of parasite or by counting parasites in one of the migratory compartment at later time point (skin, lungs or gut) with *S. ratti*?

Minor comments:

- Is the human skin treatment with micro-abrasion required for penetration ?
- It is strange that both Haloperidol and Dopamine results are presented with a unique dose given the novelty of the finding. Are the effect dose dependent ?

(Remarks on code availability)

I am not competent to review the code unfortunately.

Reviewer #2

(Remarks to the Author)

Patel et al. revealed that three species of skin-penetrating nematodes use dopamine signaling for penetration through molecular pathways similar to those already described for the recognition of food (bacterial lawn) in *C. elegans*. Specifically, they found that TRP-4, a key molecule for mechanosensation in the *C. elegans* dopaminergic neurons, is responsible for penetration. Since TRP-4 is not conserved in humans but is found in other parasitic nematodes, it is possible to develop a drug targeting TRP-4 to prevent infection by skin-penetrating nematodes.

As a specialist in *C. elegans* neurobiology but not in other nematodes, I read this manuscript with particular interest. The authors have effectively utilized the previous knowledge of *C. elegans* DA neurons and the current molecular genetic techniques in parasitic nematodes. I also found that the manuscript was very carefully written in general. I have some comments, but none of them are critical.

Minor comments:

Results

- 1) The shape of the worms was visualized using the introduced fluorescent proteins or DiI. Is it possible to demonstrate that these manipulations did not affect the worms' behavior?
- 2) The authors describe "push" and "puncture" as distinct events. However, it was not clear to me how they could distinguish between the two just by observing the video (Fig. 1 and the Movie).
- 3) Fig. 5B: I could not find the description of how the reporter constructs were made. Which part of the genomic DNA was used as the promoter, for example?
- 4) Fig. 5B: While I know how *C. elegans* DA neurons are localized, I find it difficult to confirm if they are in "the same position." I would appreciate seeing much larger images at higher magnification. Ideally, 3D movies with surrounding neurons stained with DAPI, for example. A similar comment applies to Fig. 7B.
- 5) Related to the above: Please provide information about the neurons in these parasitic worms. How many neurons are there? 302, like in *C. elegans*? Are gene expression profiles in the neurons similar to those in *C. elegans*? These may be essential if we need to assume the functions of neurons in the parasitic worms.

Discussion

- 6) The involvement of DA neurons and TRP-4 makes sense and is very interesting. However, I am not sure when and how DA signaling is involved. Reduction in dopamine signaling suppresses the skin penetration percentage, the time spent on pushing or puncturing, and the duration of pushing bouts, and extends the time to first puncture, the percentage of push and puncture followed by reversals and of aborted attempts. Can all of these effects be explained simply by a reduction in "force" for pushing due to the reduction in DA signaling, for example?

Figures

- 7) Fig. 2D: I was not able to understand why penetration frequencies are similar in *S. ratti* and *S. stercoralis*.
- 8) Fig. 3 and Fig. 4 essentially look similar. Including just one of them as the main figure may be sufficient. (Both in the main figure could also be acceptable, though.)

Methods

- 9) Single-worm skin penetration tracking assays with *S. s* and *S. r* on rat skin, 3rd paragraph: I did not understand the difference (again) between "push" and "puncture." I read the definition of "push" and "reverse," but not of "puncture."
- 10) Generation of stable mutant lines using CRISPR/Cas9-mediated mutagenesis: It was very difficult to understand what "pPRxx" plasmids are. Please briefly describe them here.
- 11) In the same paragraph: I did not understand why dual-color worms (i.e., including two transgenic genes) were required.

(Remarks on code availability)

Reviewer #3

(Remarks to the Author)

(Remarks on code availability)

Reviewer #4

(Remarks to the Author)

This manuscript, entitled "Dopamine signaling drives skin invasion by human-infective nematodes", describes investigations into the behavior and neurobiology of skin penetration by several species of important parasitic nematodes. Members of the genus *Strongyloides* are obligate skin penetrators, whereas the hookworm *Ancylostoma ceylanicum* is capable of infecting its hosts orally as well. This paper presented convincing evidence that dopamine signaling mediates skin penetration. Using fluorescently label worms and time-lapse microscopy, they showed that pharmacological inhibition of dopamine signaling inhibited behaviors involved in skin penetration. They further showed that genetic disruption of dopamine biosynthesis and chemogenetic silencing of dopaminergic neurons block penetration of both human and rat skin. This is a well-conceived and executed study and a clearly written, comprehensive manuscript that will be positively received by the parasitic nematode community, and warrants publication with the following minor revisions.

The most significant limitation of this study is the omission of experiments to determine whether the rat the parasite *S. ratti* could penetrate the skin of a non-permissive human host. Their experiments with *S. ratti* and *S. stercoralis* on rat skin

suggested a host-specific difference in skin penetrating behaviors, with *S. ratti* exhibiting more pushes on rat skin and *S. stercoralis* demonstrating more pushes on human skin. Without the converse experiment, i.e. *S. ratti* on human skin, the support for their conclusion is less rigorous. Addition of these experiments would strengthen the manuscript.

Other, less significant, issues:

Line 25: The statement that blocking skin penetration is “unexplored” is misleading. The idea of using topical chemicals to block skin penetrating organisms, including hookworms and *Strongyloides*, has been circulating for decades if not longer. It has mostly been abandoned due to the expense and impracticality of daily application of topicals by populations in endemic regions. The authors should make this less definitive.

L46: Change ineffectual to ineffective.

L48: The statement regarding anthelmintic resistance in livestock would be better supported by including more citations, including reviews, as well as recent reports of resistance in skin penetrating hookworms.

L51-53: The authors should consider that while blocking skin penetration may be a promising intervention in *Strongyloides* infections, many hookworm species are also able to infect orally, and in some cases, oral infection is the most likely transmission route.

L78-80: Change complex to complicated. From a parasitologist’s perspective, the lifecycle is not complex as it only involves one species of host. Also change this in the figure legend (Extended data figure 1).

L100: Addition of the actual number of aborted attempts would be helpful.

L113: Should specify definitive host skin.

L125: Section on pharmacological inhibition of dopamine signaling. The authors fail to specify the species of skin used in these assays. The same applies to figures 3 and 4.

L143: The phrase “drive to penetrate” is anthropomorphic. Please re-phrase to be less so.

L174-175: We find it interesting that the puncturing behavior was better in the non-definitive host tissue after the disruption of the gene. Could the authors provide any speculation as to why this might be?

L196, 258, 275: Excessive use of rhetorical questions.

L201: Isn’t a one-to-one homolog an ortholog?

L225: Remove “human” as studies were performed with non-human host skin as well.

L268-270: Is there a reason why the authors did not perform nose touch assays on iL3 of the *Ss-trp-4* knock outs? Seems like a straightforward test to confirm orthologous gene function.

Discussion: The authors should address the possible effects of skin preparation on the assays. For example, does shaving the skin introduce small cuts or nicks that alter behavior? Does the absence of hair affect penetration, possibly by removing a cue for a hair follicle? The assays seem robust, but the authors should address these and other possible weaknesses.

- Validity: The results reported are valid. One experiment would improve the manuscript.
- Originality and significance: The manuscript is original and the conclusions are valid, although they overstate the novelty of blocking skin penetration as an intervention. This paper will be of interest to the community and possibly other disciplines.
- Data & methodology: No issues.
- Appropriate use of statistics and treatment of uncertainties: There are no error bars on any of the bar graphs in Figures 2-7 or extended data figures 5, 6, or 8.
- Conclusions: No issues.
- Suggested improvements: See above.
- References: Should include additional references about anthelmintic resistance as pointed out above.
- Clarity and context: No issues.
- Inflammatory material: No issues.
- Springer Nature is committed to diversity, equity and inclusion; please raise any concerns that may in your view have an impact on this commitment. None
- Please indicate any particular part of the manuscript, data, or analyses that you feel is outside the scope of your expertise, or that you were unable to assess fully. Statistics.

(Remarks on code availability)

Reviewer #5

(Remarks to the Author)

(Remarks on code availability)

Version 1:

Reviewer comments:

Reviewer #1

(Remarks to the Author)

Many thanks for addressing all comments. The in vivo effectiveness of the treatment is in particular striking.

(Remarks on code availability)

All comments have now been replied to and I look forward to read again this manuscript.

Reviewer #2

(Remarks to the Author)

The authors have adequately addressed most of the previous reviewer comments. I believe the manuscript is acceptable for publication once the following minor issue is resolved.

In the response to Minor comments (1) of the Reviewer #2, the authors wrote:

"regardless of the slower speed and shorter distance travelled by Dil-stained iL3s on agar plates, we have shown that both Dil-stained and transgenic iL3s are equally able to penetrate skin (Fig. 2D, Fig. 3B). Moreover, both Dil-stained and transgenic iL3s spend about 20% of the time on the surface of rat skin pushing and puncturing the skin (Fig. 2B, Fig. 3C)."

However, it was not clear to me how the authors can claim that Dil-stained and transgenic worms are equally able to penetrate the skin. In fact, the methods for fluorescent labeling of the worms are not described in the figure legends for Figures 2 and 3.

(Remarks on code availability)

Reviewer #3

(Remarks to the Author)

(Remarks on code availability)

Reviewer #4

(Remarks to the Author)

This manuscript was co-reviewed with one of the reviewers who provided the listed reports. This is part of the Nature Communications initiative to facilitate training in peer review and to provide appropriate recognition for Early Career Researchers who co-review manuscripts.

The reviewers agree on the comments below. This is the review of a revised manuscript.

The authors have conclusively addressed our original comments, as well as providing extensive answers to the comments of the other reviewers. We recommend publication of the manuscript in its current form.

(Remarks on code availability)

Reviewer #5

(Remarks to the Author)

(Remarks on code availability)

Response to Reviewer Comments: Dopamine signaling drives skin invasion by human-infective nematodes

We appreciate the reviewers' enthusiasm for our manuscript. In response to their insightful comments and suggestions, we have made the changes to our manuscript that are described below.

Reviewer #1

Patel et al. describe a role for dopamine signaling in skin penetration behavior in clade V helminths. The authors suggest skin treatment as a preventative approach against skin penetrating nematode. The project is extremely original, well designed and clearly written.

Major comments:

-A major limitation of the study is the use of frozen skin. The authors state doing experiments in both fresh and frozen tissue but do not report the data to evaluate a potential difference. The problem of the dead skin is that the skin permeability will likely be affected, and one might wonder whether it then recapitulate a live infection.

We thank the reviewer for this comment. We have now included data from skin-penetration assays of *S. stercoralis* iL3s on fresh vs. frozen rat skin (Extended Data Fig. 2C). We do not see a significant difference in the proportion of iL3s that completed penetration on fresh vs. frozen skin, suggesting that iL3s were equally able to penetrate fresh rat skin and rat skin that was previously frozen and thawed.

-The rate of skin penetration reported is indeed incredibly high (80-100%) as compared to previously reported results of 20% (*S. ratti* in human skin, doi.org/10.1016/j.exppara.2015.01.005). The bar plots do not have any error bar to assess the variability of the model.

Our prior dataset did not include *S. ratti* on human skin. We now have included data showing skin-penetration behavior of *S. ratti* on human skin. In our studies, only ~40% of *S. ratti* iL3s penetrated human skin, as compared with 100% that penetrated rat skin (Fig. 2H).

The penetration rate of *S. ratti* on human skin in our studies is higher than previously reported in Jannasch *et al.*, 2015. In this paper, the authors assayed penetration by creating a chamber wherein human skin was placed over an acceptor compartment (a bath of liquid). The iL3s were then suspended in a donor compartment (another bath of liquid) on top of the human skin and allowed to invade for 8 h. After 8 h, the number of iL3s in the donor and acceptor compartments were scored and graphed. They found 80% of the iL3s in the donor compartment; based on this observation, they concluded that only 20% of the iL3s penetrated the tissue. However, in our assays where we watch iL3s penetrate skin in real-time, we find that worms that invade skin sometimes later re-emerge on the skin surface. Thus, Jannasch *et al.*, 2015 might have underestimated penetration ability of *S. ratti* iL3s on human skin, as some iL3s in their assay might have penetrated the skin but then later re-emerged into the donor compartment, appearing as worms that did not penetrate even though they did.

Please note that there are no error bars in the bar plots because the graphs show the percentages of iL3s that completed penetration on each skin type out of the total number of iL3s tested. This is now clarified in each figure legend.

-Could the authors assess the potential of Haloperidol and Dopamine treatment in an *in vivo* experiment, either by following with live imaging the entrance of parasite or by counting parasites in one of the migratory compartments at later time point (skin, lungs or gut) with *S. ratti*?

We have now tested the ability of haloperidol to block penetration by *S. ratti* using an *in vivo* skin-penetration assay, in which iL3s were placed onto the skin of live, anesthetized rats and observed under a microscope (Fig. 3E-F). Excitingly, only 10-20% of iL3s treated with haloperidol penetrated skin, whereas 80% of the control worms did so (Fig. 3E). Most iL3s that failed to penetrate also failed to puncture the skin (Fig. 3F). Thus, the inhibition of skin penetration seen with haloperidol was observed *in vivo* as well as *ex vivo*.

Minor comments:

-Is the human skin treatment with micro-abrasion required for penetration?

It appears that gentle exfoliation is required for worms to penetrate human skin within 10 minutes, as very few (<10% of the worms) penetrate skin that is not treated this way. We hypothesize that this is because we are testing forearm and breast skin, whereas iL3s are thought to typically enter hosts through the thinner skin on the top of the foot or between the toes. iL3s presumably also enter through abrasions on the bottom of the foot caused by walking barefoot outdoors. However, we have not yet been able to test this hypothesis because we have not yet been able to obtain human skin from the top of the foot.

-It is strange that both Haloperidol and Dopamine results are presented with a unique dose given the novelty of the finding. Are the effects dose dependent?

We have now added dose-response data as Extended Data Fig. 5.

Reviewer #2

Patel et al. revealed that three species of skin-penetrating nematodes use dopamine signaling for penetration through molecular pathways similar to those already described for the recognition of food (bacterial lawn) in *C. elegans*. Specifically, they found that TRP-4, a key molecule for mechanosensation in the *C. elegans* dopaminergic neurons, is responsible for penetration. Since TRP-4 is not conserved in humans but is found in other parasitic nematodes, it is possible to develop a drug targeting TRP-4 to prevent infection by skin-penetrating nematodes. As a specialist in *C. elegans* neurobiology but not in other nematodes, I read this manuscript with particular interest. The authors have effectively utilized the previous knowledge of *C. elegans* DA neurons and the current molecular genetic techniques in parasitic nematodes. I also found that the manuscript was very carefully written in general. I have some comments, but none of them are critical.

Minor comments:

Results

1) The shape of the worms was visualized using the introduced fluorescent proteins or Dil. Is it possible to demonstrate that these manipulations did not affect the worms' behavior?

We have now included measurements of speed and path length for wild-type iL3s, transgenic iL3s that express *Ss-act-2p::strmScarlet-1*, and wild-type iL3s stained with Dil (Extended Data Fig. 2A-B). We show that wild-type iL3s and transgenic iL3s move at roughly the same speeds and for the same distance. Dil-stained iL3s moved more slowly and for a shorter distance than wild-type and transgenic iL3s (Extended Data Fig. 2A-B). However, regardless of the slower speed and shorter distance travelled by Dil-stained iL3s on agar plates, we have shown that both Dil-stained and transgenic iL3s are equally able to penetrate skin (Fig. 2D, Fig. 3B). Moreover, both Dil-stained and transgenic iL3s spend about 20% of the time on the surface of rat skin pushing and puncturing the skin (Fig. 2B, Fig. 3C). Thus, although dye-staining affects speed of movement, it does not appear to affect the skin-penetration behaviors of the worms, which are the behaviors that we focus on in this paper. Additionally, we note that if Dil-staining was performed for a given experiment, all treatment groups were Dil-stained and only Dil-stained treatment groups were compared with each other; thus, we can rule out Dil-staining as a cause for the behavioral differences between treatment groups in our experiments.

2) The authors describe "push" and "puncture" as distinct events. However, it was not clear to me how they could distinguish between the two just by observing the video (Fig. 1 and the Movie).

We have now attempted to clarify the difference between a push and a puncture event by including an additional supplemental video, more detailed annotations in all the videos, and a more detailed description of the behaviors in the manuscript.

3) Fig. 5B: I could not find the description of how the reporter constructs were made. Which part of the genomic DNA was used as the promoter, for example?

We have now included a description of the promoter fragment contained in each reporter in the figure legends for the relevant figures. More detailed descriptions are provided in the Methods section.

4) Fig. 5B: While I know how *C. elegans* DA neurons are localized, I find it difficult to confirm if they are in "the same position." I would appreciate seeing much larger images at higher magnification. Ideally, 3D movies with surrounding neurons stained with DAPI, for example. A similar comment applies to Fig. 7B.

We have now included an additional supplemental figure and supplemental video that shows co-expression of the *Ss-dat-1* transcriptional reporter with the transcriptional reporter for the gene *Ss-rab-3*, which expresses in most *S. stercoralis* neurons (Extended Data Fig. 7, Movie S4). The video shows expression of both reporters throughout the volume of the worm. We used the *Ss-rab-3* transcriptional reporter in lieu of DAPI immunostaining.

We limited our imaging analysis to the *Ss-dat-1* and *Ss-rab-3* transcriptional reporters, because we have already shown that *Ss-trp-4* is co-expressed with *Ss-dat-1* (Fig. 7B). As such, we expect that the relative positions of the *Ss-dat-1* expressing neurons and the *Ss-trp-4* expressing neurons along the body of the worm are identical, and imaging just the *Ss-dat-1* transcriptional reporter would inform us about the position of the *Ss-trp-4* expressing neurons.

5) Related to the above: Please provide information about the neurons in these parasitic worms. How many neurons are there? 302, like in *C. elegans*? Are gene expression profiles in the neurons similar to those in *C. elegans*? These may be essential if we need to assume the functions of neurons in the parasitic worms.

S. stercoralis is thought to have a similar number of neurons to *C. elegans* based on its phylogenetic position in Nematoda¹. In addition, *S. stercoralis* and *C. elegans* have conserved sensory neuroanatomy based on SEM reconstructions of *S. stercoralis*²⁻⁴. Notably, the dopaminergic CEP neurons appeared to be positionally conserved based on SEM data⁴, and our genetic labeling data suggest that all of the dopaminergic neurons are found in conserved positions in *S. stercoralis*. These results are consistent both with SEM data for head sensory neurons²⁻⁴, older laser ablation studies of amphid neurons⁵⁻⁸, and our more recent genetic and functional identification of other *S. stercoralis* sensory neurons^{9,10}. We have now clarified the general conservation of sensory neuroanatomy between *S. stercoralis* and *C. elegans* in the manuscript.

Expression profiles of *S. stercoralis* neurons are not yet available, unfortunately. However, we have been able to genetically label *S. stercoralis* neurons using the putative homologs of marker genes from *C. elegans*, providing genetic as well as anatomical support for neuronal identification (in addition to this paper, see Bryant *et al.*, 2022⁹ and Banerjee *et al.*, 2025¹⁰).

Discussion

6) The involvement of DA neurons and TRP-4 makes sense and is very interesting. However, I am not sure when and how DA signaling is involved. Reduction in dopamine signaling suppresses the skin penetration percentage, the time spent on pushing or puncturing, and the duration of pushing bouts, and extends the time to first puncture, the percentage of push and puncture followed by reversals and of aborted attempts. Can all of these effects be explained simply by a reduction in "force" for pushing due to the reduction in DA signaling, for example?

We thank the reviewer for this question. We have updated the following paragraph in the Discussion section to clarify our model:

"Based on our results, we propose a model whereby the dopaminergic neurons sense topographical features of the skin surface, via the mechanotransduction channel *Ss-TRP-4*, and then release dopamine. Dopamine binds to downstream dopamine receptors, causing the worm to slow forward crawling and instead push against the skin; pushing behavior likely enables worms to identify entry points into the skin. Consistent with the model that dopamine and *Ss-TRP-4* are necessary for pushes, we show that *Ss-cat-2*^{-/-} and *Ss-trp-4*^{-/-} iL3s push and puncture skin for less time than wild-type iL3s (Fig. 5H-I, Extended Data Fig. 8C-D, Fig. 7G-H, Extended Data Fig. 11C-D). Continued pushing, coupled with the secretion of metalloproteases that help break down the skin,

leads to skin penetration. Our results also suggest that dopamine signaling suppresses mechanosensory behaviors that would otherwise prevent skin penetration. *Ss-cat-2^{-/-}* iL3s often reversed after a push or puncture, whereas wild-type iL3s did not (Fig. 5K); these reversals prevented the forward locomotion required for skin penetration. The increased tendency of *Ss-cat-2^{-/-}* iL3s to reverse after pushes and punctures might reflect a hypersensitivity of the mutants to nose touch, as both pushes and punctures are always preceded by head-on collisions with the skin surface. This model is consistent with studies that have shown that dopamine signaling modulates the sensitivity of *C. elegans* to stimuli that cause reversal behavior, including nose touch. *Ss-TRP-4* may mediate the effect of dopamine signaling on nose-touch sensitivity, as *Ss-trp-4* mutants also reverse frequently after a push or puncture (Fig. 7J). The dopamine signaling pathway could act constitutively or in a context-dependent manner (e.g., when the worm is on skin) to modulate nose-touch sensitivity."

Thus, we postulate that dopamine signaling controls several aspects of skin penetration. Additionally, we think it unlikely that the effect of the DA neurons on skin penetration is simply in generating the force necessary for pushing and burrowing, because the number of push attempts is lower in haloperidol-treated, DA neuron-silenced, *Ss-cat-2^{-/-}*, and *Ss-trp-4^{-/-}* iL3s.

Figures

7) Fig. 2D: I was not able to understand why penetration frequencies are similar in *S. ratti* and *S. stercoralis*.

We have now updated the manuscript and removed the data comparing the penetration frequencies of *S. ratti* and *S. stercoralis* on rat skin. However, we do find that 100% of *S. stercoralis* and *S. ratti* iL3s were able to penetrate rat skin, despite the relatively fewer pushes and punctures performed by *S. stercoralis* on rat skin. One potential reason for this is that rat skin is softer than human skin¹¹ and thus might be a relatively easy substrate to penetrate.

8) Fig. 3 and Fig. 4 essentially look similar. Including just one of them as the main figure may be sufficient. (Both in the main figure could also be acceptable, though.)

We thank the reviewer for this comment. However, we would prefer to keep Figures 3 and 4 as main figures because we are testing the effect of haloperidol on two distinct species of nematodes that occupy two different phylogenetic clades: *S. stercoralis* (clade IV) in Figure 3 and *Ancylostoma ceylanicum* (clade V) in Figure 4. *S. stercoralis* and *A. ceylanicum* are both thought to have independently evolved from a common free-living ancestor¹²⁻¹⁴ and thus are also thought to have independently evolved the ability to penetrate skin. Our data show that although both nematodes likely evolved this skin-penetration ability independently, they seem to have co-opted dopamine signaling for this purpose. We think the significance of this finding merits including both datasets in the main manuscript.

Methods

9) Single-worm skin penetration tracking assays with *S. s* and *S. r* on rat skin, 3rd paragraph: I did not understand the difference (again) between "push" and "puncture." I read the definition of "push" and "reverse," but not of "puncture."

We have now updated this section of the manuscript to include our definition of puncture events.

10) Generation of stable mutant lines using CRISPR/Cas9-mediated mutagenesis: It was very difficult to understand what "pPRxx" plasmids are. Please briefly describe them here.

We have now added a description of these plasmids to this paragraph.

11) In the same paragraph: I did not understand why dual-color worms (i.e., including two transgenic genes) were required.

We have now clarified why we prefer to use dual-colored worms for our assays and strain maintenance. Briefly, this is to ensure that we are testing homozygous mutants.

Reviewer #3

Reviewer #4

This manuscript, entitled “Dopamine signaling drives skin invasion by human-infective nematodes”, describes investigations into the behavior and neurobiology of skin penetration by several species of important parasitic nematodes. Members of the genus *Strongyloides* are obligate skin penetrators, whereas the hookworm *Ancylostoma ceylanicum* is capable of infecting its hosts orally as well. This paper presented convincing evidence that dopamine signaling mediates skin penetration. Using fluorescently label worms and time-lapse microscopy, they showed that pharmacological inhibition of dopamine signaling inhibited behaviors involved in skin penetration. They further showed that genetic disruption of dopamine biosynthesis and chemogenetic silencing of dopaminergic neurons block penetration of both human and rat skin. This is a well-conceived and executed study and a clearly written, comprehensive manuscript that will be positively received by the parasitic nematode community and warrants publication with the following minor revisions. The most significant limitation of this study is the omission of experiments to determine whether the rat parasite *S. ratti* could penetrate the skin of a non-permissive human host. Their experiments with *S. ratti* and *S. stercoralis* on rat skin suggested a host-specific difference in skin penetrating behaviors, with *S. ratti* exhibiting more pushes on rat skin and *S. stercoralis* demonstrating more pushes on human skin. Without the converse experiment, i.e. *S. ratti* on human skin, the support for their conclusion is less rigorous. Addition of these experiments would strengthen the manuscript.

We thank the reviewer for this suggestion. We have now added this experiment to the manuscript (Fig. 2E-H).

Other, less significant, issues:

Line 25: The statement that blocking skin penetration is “unexplored” is misleading. The idea of using topical chemicals to block skin penetrating organisms, including hookworms and *Strongyloides*, has been circulating for decades if not longer. It has mostly been abandoned due to the expense and impracticality of daily application of topicals by populations in endemic regions. The authors should make this less definitive.

We have changed this phrasing as suggested.

L46: Change ineffectual to ineffective.

We have made this change.

L48: The statement regarding anthelmintic resistance in livestock would be better supported by including more citations, including reviews, as well as recent reports of resistance in skin penetrating hookworms.

We appreciate this request and have now added additional citations, including reviews and reports of anthelmintic resistance in hookworms.

L51-53: The authors should consider that while blocking skin penetration may be a promising intervention in *Strongyloides* infections, many hookworm species are also able to infect orally, and in some cases, oral infection is the most likely transmission route.

This is an important point, and we have now mentioned this in the Discussion section.

L78-80: Change complex to complicated. From a parasitologist’s perspective, the lifecycle is not complex as it only involves one species of host. Also change this in the figure legend (Extended data figure 1).

We have made these changes.

L100: Addition of the actual number of aborted attempts would be helpful.

We have now added these data.

L113: Should specify definitive host skin.

We have rewritten this section to include new data. However, we have now specified that humans are a definitive host for *S. stercoralis* in this section.

L125: Section on pharmacological inhibition of dopamine signaling. The authors fail to specify the species of skin used in these assays. The same applies to figures 3 and 4.

We have now specified that rat skin was used in these assays.

L143: The phrase “drive to penetrate” is anthropomorphic. Please rephrase to be less so.

We have now replaced “skin-penetration drive” or “drive to penetrate” in the Results and Figure Legends sections with “skin-penetration behavior.”

L174-175: We find it interesting that the puncturing behavior was better in the non-definitive host tissue after the disruption of the gene. Could the authors provide any speculation as to why this might be?

We have now updated the corresponding Results section to speculate why the inhibition of punctures was less severe on rat skin than human skin. Briefly, we think this might be because rat skin is softer¹¹ and therefore likely easier to puncture and penetrate than human skin.

L196, 258, 275: Excessive use of rhetorical questions.

We have now rephrased these statements.

L201: Isn't a one-to-one homolog an ortholog?

We have updated this to change “one-to-one homolog” to “putative ortholog”.

L225: Remove “human” as studies were performed with non-human host skin as well.

We have changed the word “human” to “mammalian.”

L268-270: Is there a reason why the authors did not perform nose touch assays on iL3 of the *Ss-trp-4* knock outs? Seems like a straightforward test to confirm orthologous gene function.

This is a good suggestion. We have tested the response of *Ss-trp-4* and *Ss-cat-2* mutants to gentle nose touch and compared the responses to those of wild type. The results of these assays are shown below.

Methods and figure legends: Between 4-5 worms of each genotype were placed onto 6 cm NGM plates that had been freshly seeded with OP50 the night before. Worms were allowed to crawl on the plate for 1-2 min. Then, an eyelash was placed in the path of a forward-moving worm and the response to head-on collisions with the eyelash was recorded. The worm was considered to have responded to the collision if it reversed or turned away; continuous forward movement was considered a lack of response. Each collision was considered a single trial, and each worm was assayed for a total of 5 trials, with a rest period of 30 s between each trial. The graph on the left shows the percentage of touches that produced a response. Each dot represents a single worm; dashed and dotted lines indicate the median and interquartile ranges, respectively. n = 14-15 iL3s per condition, collected from two independent biological replicates. *** $p < 0.001$, **** $p < 0.0001$, Kruskal-Wallis test with Dunn's correction.

Results: Both *Ss-cat-2* and *Ss-trp-4* mutants show increased responsiveness to eyelash collisions, as compared with wild-type iL3s, suggesting that the mutants are hypersensitive to touch. These data are consistent with our model that dopamine signaling normally inhibits mechanosensory behaviors that might prevent skin penetration. However, we would like to follow up on these findings more thoroughly before publishing, because preliminary observations indicate that there are periods when both *Ss-cat-2* and *Ss-trp-4* are more responsive to nose touch and periods when they are not. The periods of increased responsiveness seem to occur right after *Ss-cat-2* and *Ss-trp-4* undergo spontaneous reversals. Additionally, because iL3s are very skinny (16 μm in diameter), we were not always sure if the worms fully contacted the eyelash or if they simply crawled underneath it. Thus, we would like to repeat these gentle touch assays with a different device that fully contacts the head of the worm. However, this will require further assay development.

Discussion: The authors should address the possible effects of skin preparation on the assays. For example, does shaving the skin introduce small cuts or nicks that alter behavior? Does the absence of hair affect penetration, possibly by removing a cue for a hair follicle? The assays seem robust, but the authors should address these and other possible weaknesses.

We thank the reviewers for this comment and have now included a paragraph in the Discussion section that discusses these questions.

- Validity: The results reported are valid. One experiment would improve the manuscript.

We have now added this experiment.

- Originality and significance: The manuscript is original, and the conclusions are valid, although they overstate the novelty of blocking skin penetration as an intervention. This paper will be of interest to the community and possibly other disciplines.

We have rephrased the manuscript accordingly.

- Data & methodology: No issues.

- Appropriate use of statistics and treatment of uncertainties: There are no error bars on any of the bar graphs in Figures 2-7 or extended data figures 5, 6, or 8.

We have now updated the figure legends to specify why there are no error bars on the bar graphs. This is because these graphs show the percentage of iL3s that penetrated skin out of the total number of iL3s tested, across all trials (and the corresponding data were analyzed using Fisher's exact test).

- Conclusions: No issues.
- Suggested improvements: See above.
- References: Should include additional references about anthelmintic resistance as pointed out above.

Additional references have been added.

References

- 1 Schafer, W. Nematode nervous systems. *Curr Biol* **26**, R955-R959, doi:10.1016/j.cub.2016.07.044 (2016).
- 2 Ashton, F. T., Bhopale, V. M., Fine, A. E. & Schad, G. A. Sensory neuroanatomy of a skin-penetrating nematode parasite: *Strongyloides stercoralis*. I. Amphidial neurons. *J Comp Neurol* **357**, 281-295, doi:10.1002/cne.903570208 (1995).
- 3 Ashton, F. T. & Schad, G. A. Amphids in *Strongyloides stercoralis* and other parasitic nematodes. *Parasitol Today* **12**, 187-194 (1996).
- 4 Fine, A. E., Ashton, F. T., Bhopale, V. M. & Schad, G. A. Sensory neuroanatomy of a skin-penetrating nematode parasite *Strongyloides stercoralis*. II. Labial and cephalic neurons. *J Comp Neurol* **389**, 212-223 (1997).
- 5 Ashton, F. T., Bhopale, V. M., Holt, D., Smith, G. & Schad, G. A. Developmental switching in the parasitic nematode *Strongyloides stercoralis* is controlled by the ASF and ASI amphidial neurons. *J Parasitol* **84**, 691-695 (1998).
- 6 Ashton, F. T., Zhu, X., Boston, R., Lok, J. B. & Schad, G. A. *Strongyloides stercoralis*: amphidial neuron pair ASJ triggers significant resumption of development by infective larvae under host-mimicking *in vitro* conditions. *Exp Parasitol* **115**, 92-97, doi:10.1016/j.exppara.2006.08.010 (2007).
- 7 Forbes, W. M., Ashton, F. T., Boston, R., Zhu, X. & Schad, G. A. Chemoattraction and chemorepulsion of *Strongyloides stercoralis* infective larvae on a sodium chloride gradient is mediated by amphidial neuron pairs ASE and ASH, respectively. *Vet Parasitol* **120**, 189-198, doi:10.1016/j.vetpar.2004.01.005 (2004).
- 8 Lopez, P. M., Boston, R., Ashton, F. T. & Schad, G. A. The neurons of class ALD mediate thermotaxis in the parasitic nematode, *Strongyloides stercoralis*. *Int J Parasitol* **30**, 1115-1121, doi:10.1016/s0020-7519(00)00087-4 (2000).
- 9 Bryant, A. S., Ruiz, F., Lee, J. & Hallem, E. A. The neural basis of heat seeking in a human-infective parasitic worm. *Curr Biol* **32**, 2206-2221, doi:10.1101/2021.06.23.449647 (2022).
- 10 Banerjee, N. *et al.* Carbon dioxide shapes parasite-host interactions in a human-infective nematode. *Curr Biol* **35**, 277-286.e276, doi:10.1016/j.cub.2024.11.036 (2025).
- 11 Wei, J. C. J. *et al.* Allometric scaling of skin thickness, elasticity, viscoelasticity to mass for micro-medical device translation: from mice, rats, rabbits, pigs to humans. *Sci Rep* **7**, 15885, doi:10.1038/s41598-017-15830-7 (2017).
- 12 Blaxter, M. L. *et al.* A molecular evolutionary framework for the phylum Nematoda. *Nature* **392**, 71-75 (1998).
- 13 Blaxter, M. & Koutsovoulos, G. The evolution of parasitism in Nematoda. *Parasitol* **142 Suppl 1**, S26-S39 (2015).
- 14 Blaxter, M., Koutsovoulos, G., Jones, M., Kumar, S. & Elsworth, B. in *Next Generation Systematics* (eds P.D. Olson, J. Hughes, & J.A. Cotton) 62-82 (Cambridge University Press, 2016).

Response to Reviewer Comments:
Dopamine signaling drives skin invasion by human-infective nematodes

We were happy to see that we had addressed the reviewer's previous comments and concerns. Reviewer #2 had one remaining minor comment, which we have addressed below.

Reviewer #2

The authors have adequately addressed most of the previous reviewer comments. I believe the manuscript is acceptable for publication once the following minor issue is resolved.

In the response to Minor comments (1) of the Reviewer #2, the authors wrote: "regardless of the slower speed and shorter distance travelled by Dil-stained iL3s on agar plates, we have shown that both Dil-stained and transgenic iL3s are equally able to penetrate skin (Fig. 2D, Fig. 3B). Moreover, both Dil-stained and transgenic iL3s spend about 20% of the time on the surface of rat skin pushing and puncturing the skin (Fig. 2B, Fig. 3C)."

However, it was not clear to me how the authors can claim that Dil-stained and transgenic worms are equally able to penetrate the skin. In fact, the methods for fluorescent labeling of the worms are not described in the figure legends for Figures 2 and 3.

We have now clarified that both Dil-stained and transgenic iL3s show similar behaviors on the skin surface. These behaviors are quantitatively as well as qualitatively similar – both groups of iL3s show spend the same percentage of time pushing and puncturing the skin surface. The only difference is that the Dil-stained worms crawl at a slightly reduced speed.

In our manuscript, we only ever compare Dil-stained worms to other Dil stained worms, and transgenic worms to other transgenic worms. In addition, the metrics we quantify do not include speed. Thus, the slightly reduced speed of Dil-stained worms does not affect any of the metrics reported in the manuscript.

We have now clarified this point in the manuscript. We have also clarified that transgenic iL3s were used in Fig. 2 and Dil-stained iL3s were used in Fig. 3.